# Colossal thermo-hydro-electrochemical voltage generation for self-sustainable operation of electronics

Yufan Zhang[1], Ahrum Sohn[2], Anirban Chakraborty[2] & Choongho Yu [1,2]✉

Thermoelectrics are suited to converting dissipated heat into electricity for operating electronics, but the small voltage (~0.1 mV K⁻¹) from the Seebeck effect has been one of the major hurdles in practical implementation. Here an approach with thermo-hydro-electrochemical effects can generate a large thermal-to-electrical energy conversion factor (TtoE factor), −87 mV K⁻¹ with low-cost carbon steel electrodes and a solid-state polyelectrolyte made of polyaniline and polystyrene sulfonate (PANI:PSS). We discovered that the thermo-diffusion of water in PANI:PSS under a temperature gradient induced less (or more) water on the hotter (or colder) side, raising (or lowering) the corrosion overpotential in the hotter (or colder) side and thereby generating output power between the electrodes. Our findings are expected to facilitate subsequent research for further increasing the TtoE factor and utilizing dissipated thermal energy.

[1] Department of Materials Science and Engineering, Texas A&M University, College Station, Texas 77843, USA. [2] Department of Mechanical Engineering, Texas A&M University, College Station, Texas 77843, USA. ✉email: chyu@tamu.edu

Thermoelectricity refers to converting heat to electricity or vice versa, and has been used for various applications, including thermocouples and Peltier devices. It is desired to generate a large voltage per temperature difference (i.e., large thermopower or Seebeck coefficient in the unit of $V\,K^{-1}$), resulting from the thermodiffusion of electrons called the Seebeck effect[1]. As for traditional solid-state inorganic and recent organic thermoelectric materials, small thermopower values on the order of $0.01–0.1\,mV\,K^{-1}$ near room temperature have been one of the major hurdles in acquiring high device performances[2–7]. In the last several years, the thermodiffusion of ions, called the Soret effect[8], has been utilized to aggrandize thermopower to $8\,mV\,K^{-1}$ with polystyrene sulfonic acid due to the substantial difference in thermodiffusion between mobile protons and immobile anions[9]. While it has been difficult to have greater than several $mV\,K^{-1}$, several recent studies yielded gigantic thermopower[10–13] including $18\,mV\,K^{-1}$ from the migration of chloride ions in n-type mixed ionic–electronic conductive polymer composite films[14], $17\,mV\,K^{-1}$ partially from KCl in a quasi-solid-state ionic thermoelectric material[15], and $24\,mV\,K^{-1}$ with sodium ions in cellulose ionic conductors[16]. It is worth noting that the thermally induced voltage is different from moisture-powered generators (MPG) based on ion movement carried by water diffusion and ion accumulation[17–23]. MPG works only when environmental humidity changes or water droplets are applied to the device, and it does not respond to a temperature difference.

To induce a thermally induced voltage on the order of $1–10\,mV\,K^{-1}$, a few different mechanisms have been recently reported, including the thermodiffusion of electron/ion mixture (both Soret and Seebeck effects)[24,25] and temperature-dependent redox reactions with redox couples in liquid states[26–28]. It appears that solid-state polyelectrolytes utilizing the Soret effect are the best option for high thermopower so far, and their highest thermopower values were obtained at unusually high (70–100%) relative humidity (RH) rather than typical room humidity (~50% RH). Water is an electrolyte for the ions, improving their mobility, and water makes mobile ions readily dissociated from their counter ions[9,12,24]. However, high water uptake in solid-state polyelectrolytes often causes stability problems due to irreversible water evaporation and swelling. Considering that high thermal-to-electrical conversion (TtoE) factor is the key to the performance, it is valuable to seek other routes for attaining even bigger TtoE factors. It should be noted that we used a term TtoE factor rather than thermopower and Seebeck coefficient in order to simultaneously account for various principles generating thermally induced voltage. For example, to acquire a working voltage (>1 V) of typical wearable electronics with traditional inorganic materials, at least 1000 thermoelectric legs should be serially connected under a temperature difference of 10 °C.

Here we report an approach based on the variation of potentials caused by a temperature difference. We used readily available carbon steel as electrodes to obtain a colossal TtoE factor of $-87\,mV\,K^{-1}$ under a typical ambient condition (50% RH, 22 °C). Porous hydrophilic layers were formed on the carbon steel, and a hygroscopic solid-state polyelectrolyte layer was used between the two electrodes. We discovered that, upon imposing a temperature difference, the thermodiffusion of water from the hotter side to the colder side altered the water uptake in the electrode, differentiating the potential of the two electrodes. Based on the mechanism, a self-sustainable fever-detection device has been operated as a proof of principle, which could be helpful in the early and fast detection of fever commonly observed from a viral infection such as COVID, SARS, MERS, and swine flu pandemic. The following include fabrication and characterization of materials and devices, investigation of working principles, and applications.

## Results and discussion

Our device consists of polyaniline and polystyrene sulfonate (PANI:PSS) as a solid-state electrolyte and carbon steel foils as electrodes (Fig. 1a). PANI:PSS powders (see Supplementary Fig. 1 for FTIR) were synthesized with polystyrene sulfonic acid (PSS-H) and aniline[29], and then they were dissolved in deionized water with hydrochloric acid. The solution was drop-casted on two carbon steel electrodes, and two pieces were assembled before they were fully dried. After the assembly, the sample was left in a fume hood. During this time period, the surface of the carbon steel was corroded, forming a new layer between PANI:PSS and electrodes, as shown in Fig. 1b. We observed a porous layer composed of few-micron-long nanorods (Fig. 1c) under a flat PANI:PSS layer (Fig. 1d). The XRD patterns (Fig. 1e) indicate that the nanorod is made of $\beta$-FeOOH[30]. As PANI:PSS is a hygroscopic material[9,31], the water uptake is a strong function of RH in the environment. The amount of water soaked in PANI:PSS under different RH was characterized as a function of time (see Supplementary Fig. 2), and steady-state values are summarized in Fig. 1f. The water uptake in the sample was augmented with higher RH, and was found to be ~15 wt% in PANI:PSS under a typical room environment (50% RH). Transport-property measurements were carried out after the water-uptake-reached steady states.

For thermally induced voltage measurements, the temperature difference between two electrodes was varied up to ±6 K, and voltage was recorded as a function of time. Figure 2a shows the generated voltage from the devices with 15% water uptake (RH = 50%), and the saturated voltage as a function of temperature difference was plotted in the inset of Fig. 2a. Those of all the other samples are shown in Supplementary Fig. 3, Supplementary Table 1, and Supplementary Table 2. The slopes from the linear fitting are the absolute values of the TtoE factor, which are plotted against RH along with those from various thermal-to-electrical energy-conversion principles in the literature (Fig. 2b)[9,10,12–15,24,32–34]. It should be noted that the sign of the number for the slope should be reversed to get the TtoE factor (i.e., a positive slope means a negative TtoE factor) like conventional thermopower. We found that the magnitude of the TtoE factor got bigger as we elongated the oxidation time of the carbon steel in the ambient condition, but it did not further increase after ~60 days. We observed consistent values, $-85 \sim -87\,mV\,K^{-1}$ after 120 days and 180 days. The two different cases from fully developed (60 days) and intermediate (14 days) oxidation layers are shown in Fig. 2b. At 30% RH, the TtoE factor of the fully developed case was $-48\,mV\,K^{-1}$, and, under 50% RH, it was boosted up to $-87\,mV\,K^{-1}$. Even for the intermediate case, $-47\,mV\,K^{-1}$ at 50% RH in our thermo–hydro-electrochemical hybrid device is much higher than other thermally induced voltages in the literature.

The difference in the TtoE factor for the two cases mainly comes from the impedance change of the oxidation layer. A greater potential difference between electrodes can be developed when the impedance of the oxidation layer was enlarged. According to the electrochemical impedance spectroscopy results (see Supplementary Fig. 4), the impedance of PANI:PSS on the order of 10 Ω was significantly raised to values on the order of kΩ with the oxidation layers, and larger impedance was observed from the fully developed cases. It is interesting to see the distinct humidity dependency from our sample where there is an optimum RH, while the others show monotonically increasing TtoE factors with RH. In fact, the optimum performance at 50% RH is ideal because it is close to that of typical indoor environments. On the other hand, this would be an indicator that the working principle of our system is different from the others.

In the literature reporting high TtoE factors, the thermodiffusion of ions (e.g., proton) was identified to be the main

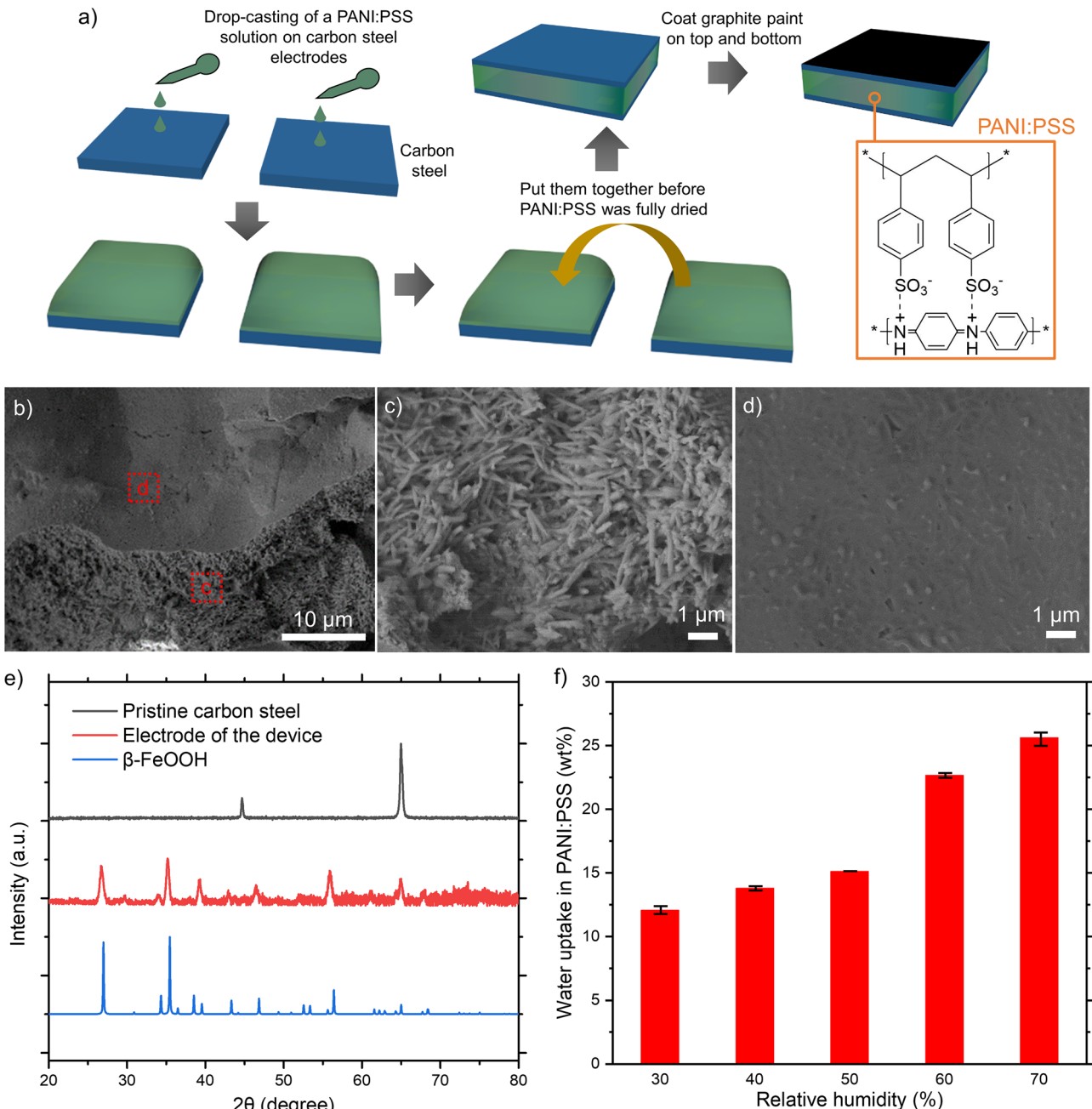

**Fig. 1 Device fabrication and characterization. a** Device-fabrication procedure. Scanning electron micrographs of (**b**) the carbon steel electrode after detaching PANI:PSS, **c** a porous area, and (**d**) a flat area. **e** X-ray diffraction data for a pristine carbon steel, the electrode shown in "b", and reference data for β-FeOOH[30]. **f** Water uptake of PANI:PSS after the sample was left under the relative humidity for 8 h. Error bar: one standard deviation.

driver[8,9,11–16,35,36]. Here, to test the influence of the thermo-diffusion of ions in PANI:PSS on the TtoE factor, we assembled a cell with graphite foils instead of carbon steel (see Supplementary Fig. 5a). We found that the slope in temperature difference vs. voltage plot in Supplementary Fig. 5b is opposite to that of our device in the inset of Fig. 2a. When protons in PANI:PSS migrate from the hotter side to the colder side, a negative slope (or a positive TtoE factor) was obtained. Moreover, the TtoE factor from the device with graphite electrodes was found to be only 0.97 mV K$^{-1}$, which is much smaller than −87 mV K$^{-1}$ from our carbon steel-based device. Therefore it is clear that the working principle of our device is different from those in the literature. Instead, the graphite device is similar to MPG[17–23] because they both rely on proton migration. To verify the difference between

MPG and ours, we carried out experiments exposing moisture instead of heat to one of the electrodes (see Supplementary Fig. 6). PANI:PSS with our carbon steel electrodes generated ~360 mV under RH difference of 58%, whereas the graphite-electrode device generated only ~0.2 mV. We also tested the influence of β-FeOOH/Fe$^{2+}$ redox couple on the TtoE factor by sandwiching β-FeOOH between PANI/PSS and graphite foils. We observed that voltage continuously changed, despite the constant temperature difference with an absolute value of ~1.3 mV at most under 4 K difference (see Supplementary Fig. 7). The large TtoE factor can be obtained only with the steel electrodes, suggesting that a corrosion process plays a key role in our system.

Then we investigated if corrosion was caused by the thermo-diffusion of protons in a 3-electrode configuration with the

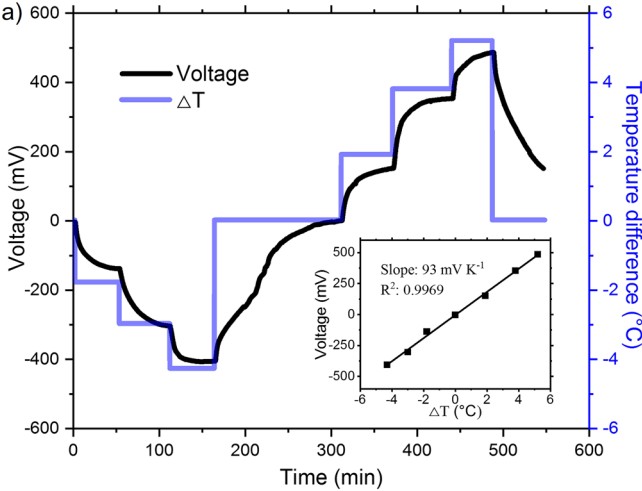

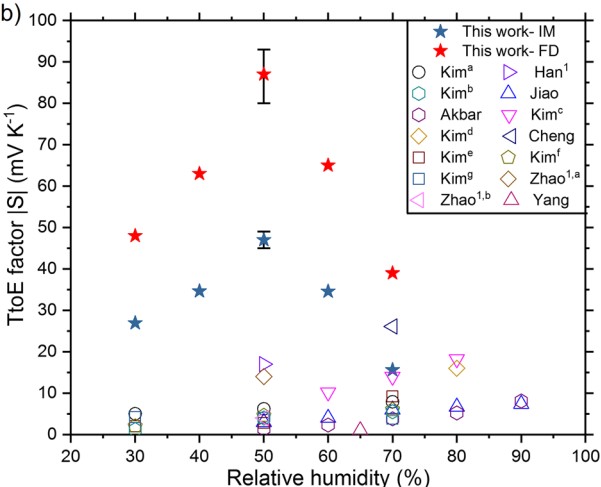

**Fig. 2 Thermal-to-electrical conversion factor. a** An exemplary voltage profile of the device with the fully developed oxidation layer under 50% RH as a function of time when the temperature difference (ΔT) was altered. The inset shows the saturated voltage at the corresponding temperature to seek the slope (93 mV K$^{-1}$). Note that the TtoE factor has the opposite sign ($-93$ mV K$^{-1}$) like conventional thermopower. **b** TtoE factors (absolute value) of our devices with the intermediate (IM) and fully developed (FD) oxidation layers along with data in the literature, Kim[a 9], Han[15], Kim[b 12], Jiao[13], Akbar[10], Kim[c 14], Kim[d 24], Cheng[32], Kim[e 11], Kim[f 11], Kim[g 12], Zhao[a 33], Zhao[b 33], and Yang[34]. Error bar: one standard deviation. [1]RH is assumed to be 50%.

FeOOH-coated carbon steel electrode (taken out from the device) as a working electrode (see Supplementary Fig. 8). Voltage sweeping between a Pt counter and reference electrodes resulted in Tafel curves, showing a corrosion potential of $-0.45$ V vs. Ag/AgCl when the current between the working and counter electrodes approached zero. As HCl was gradually added to have pH of 1, 0.8, and 0.6 (initially pH = 1.25), the potential shifted toward positive values with more protons (or a lower pH). This result denotes that more protons make the corrosion potential of carbon steel more positive, which is the same as Supplementary Fig. 5 where negative voltages under ΔT > 0 (protons on the bottom electrode) were observed. This behavior is opposite to the trend of our device in Fig. 2a. A similar experiment (see Supplementary Fig. 9) also exhibited that voltage was shifted more positively with the addition of protons, confirming that the thermodiffusion of the proton is not the major contributor in our

system. Furthermore, our experimental results in Supplementary Fig. 10 proved that the corrosion potential in the Tafel curves is not a significant function of temperature.

As voltage generation strongly depends on humidity, we carried out experiments directly showing the influence of water uptake on voltage generation. One of the electrodes in the device was taken out of the device and exposed to environmental conditions whose RH was altered from 50% RH to 20% RH and 70% RH for 12 h (Fig. 3a). The porous layer on the electrode can accommodate water from the humid environment or release water initially present in the 50% RH condition, as indicated by the mass change in drier and wetter conditions (Fig. 3b). When the electrode was reassembled, the voltage was remarkably altered, showing higher (or lower) potential with less (or more) water. When water moves from the hotter side to the colder side, the hotter side has a higher potential than the colder side, which agrees with the trend (slope) shown in our system (inset of Fig. 2a).

We used in situ attenuated total reflectance (ATR) Fourier transform-infrared spectroscopy (FTIR) to identify the thermo-diffusion of water in PANI:PSS by comparing the intensity of O–H stretching band for water,[37] which appears over a broad range near 2800–3700 cm$^{-1}$ with its peak[38] at ~3450 cm$^{-1}$ while one side of PANI:PSS was being heated (Fig. 3c). All the spectra were normalized by the peak intensity at 2914 cm$^{-1}$ corresponding to C–H stretching of the CH$_2$ group in PSS[39]. The absorbance peak near 3450 cm$^{-1}$ from the colder side was intensified as the temperature difference was enlarged, suggesting that water diffused from the hotter to the colder side. It should be noted that the colder side of the sample was in contact with the FTIR apparatus at room temperature to avoid water condensation, and a temperature lower than room temperature was not applied due to the risk of changes in the water content.

The Evans diagram in Fig. 3d explains how the voltage was developed. Initially, the corrosion potential is located at the intersection between the oxidation and reduction Tafel curves (greenish and black lines in Fig. 3d, respectively). Under a temperature gradient, the thermodiffusion of water reduces the amount of water in the hotter side, while that in the colder side increases. Water is an electrolyte in corrosion reactions, so a reduction in water renders the corrosion overpotential higher. Then the Tafel curve for oxidation shifts counterclockwise (red line in Fig. 3d), and then a new potential (crossover point) is established due to the following relation[40]:

$$E^0_i = E_{anode} + I_i R_i \tag{1}$$

$E^0_i$ and $E_{anode}$ are the potential of corrosion and anodic reaction, respectively. $I_i R_i$ is the overpotential. $I_i$ is the corrosion current and $R_i$ is the resistance of the electrolyte, where the index i is either the hotter (h) or colder side (c). On the other hand, higher water uptake decreases the overpotential, lowering the crossover point (blue line in Fig. 3d). The newly established two crossover points between the reduction line and the raised/lowered oxidation lines for the hotter/colder sides create a potential difference between the two electrodes, as follows:

$$\Delta E = E^0_h - E^0_c = I_h R_h - I_c R_c \tag{2}$$

The potential difference is a function of the corrosion current and resistance, which are strongly affected by the amount of water in the electrodes[41].

Figure 3e, f illustrates water diffusion and the corresponding electrode potentials. Under a uniform temperature, water is homogeneously distributed in PANI:PSS, and the potentials of the top and bottom electrodes are identical (Fig. 3e). When a temperature gradient is created, the water molecules in the hotter side diffuse to the colder side (Fig. 3f). As the amount of migrated

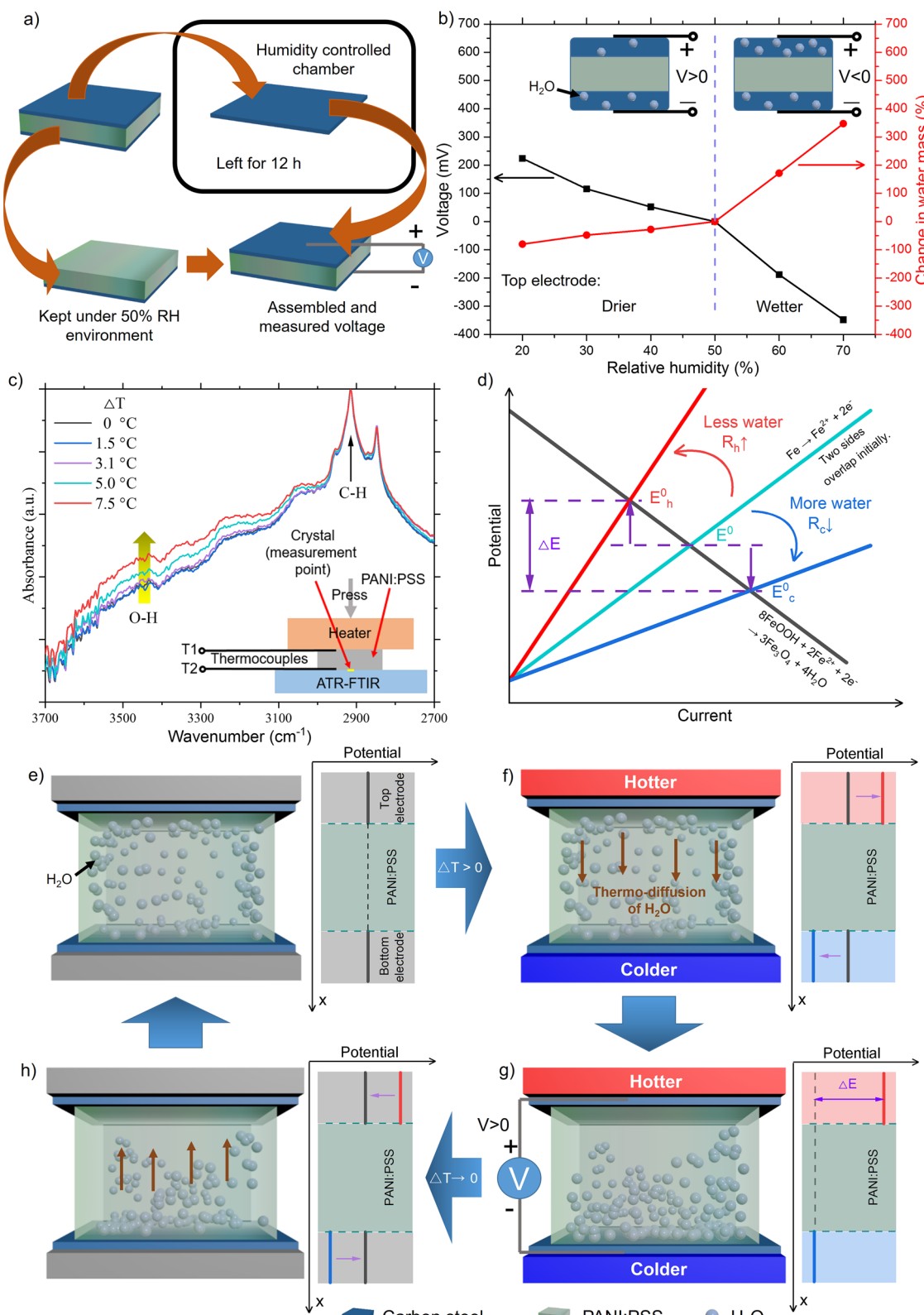

**Fig. 3 Potential change with water uptake and working principle of voltage generation. a** Experimental procedure to identify the influence of water uptake on the corrosion potential of carbon steel. **b** Voltage (left axis) and change of water mass (right axis) when the top electrode was left under different RH following the procedure in "a". **c** ATR–FTIR spectra of PANI:PSS from the colder side as the temperature difference (ΔT) was varied. The inset illustrates the in situ ATR–FTIR experiment configuration. **d** Evans diagram shows the cathodic-reduction reaction (black line), the anodic oxidation reaction (greenish line) under uniform temperature. The greenish line shifts toward the red (or blue) line with less (or more) water on the electrodes. **e–h** Illustration showing thermoelectric voltage generation and the corresponding potential changes in the hotter and colder sides. **e** Uniformly distributed water in PANI:PSS under ΔT = 0. **f** As ΔT > 0, water molecules migrate from the hotter to the colder side. **g** Voltage generation at steady state under ΔT > 0. **h** Water molecules return to their initial distributed states under ΔT→0.

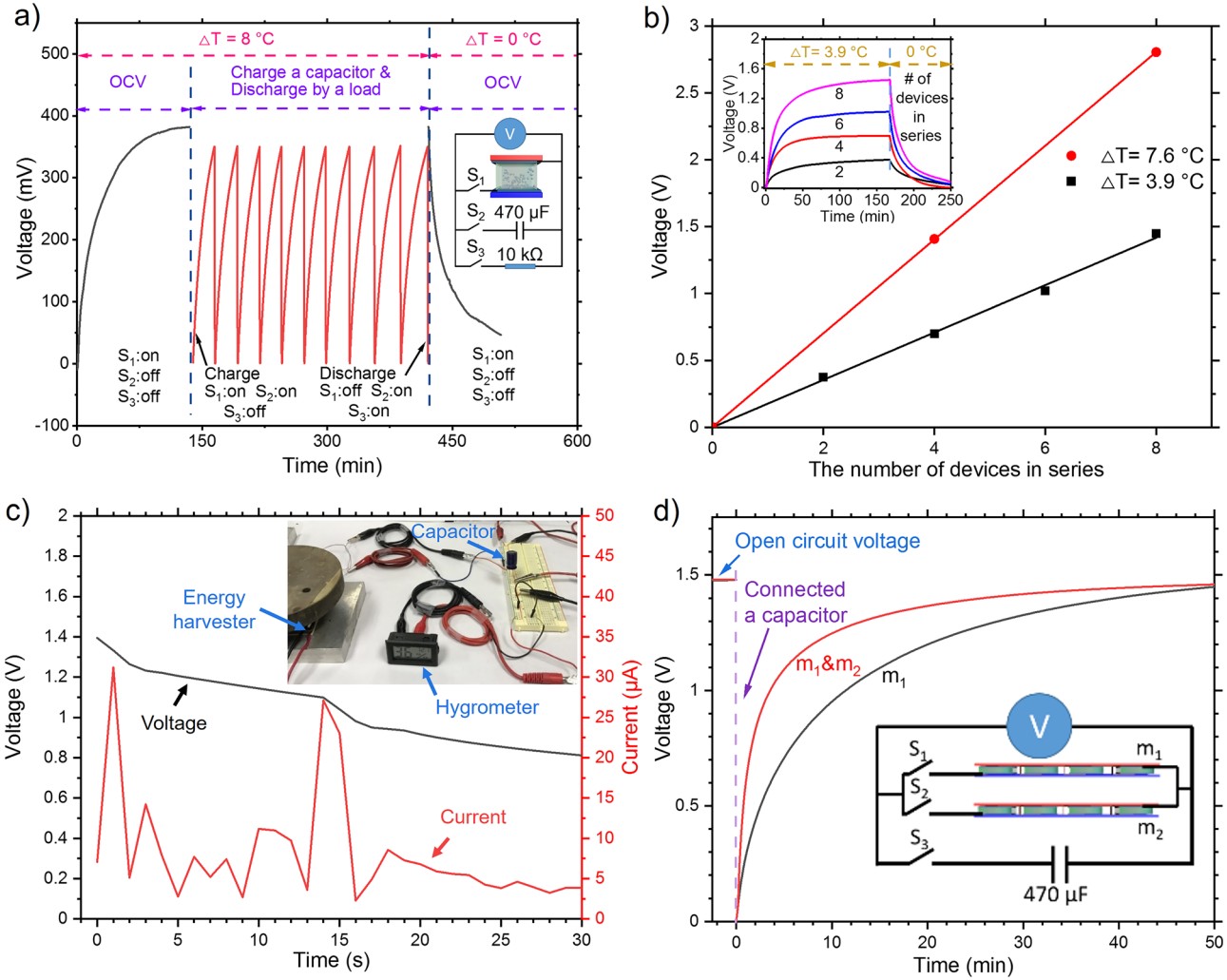

**Fig. 4 Output voltage and current with multiple devices connected in series and parallel. a** Voltage generation from one device under ΔT = 8 °C followed by charging/discharging a capacitor. **b** Linearly increasing voltage with more number of devices connected in series under ΔT = 3.9 and 7.6 °C. The upper inset is the voltage profile as a function of time as 2, 4, 6, and 8 devices were joined in series. The lower inset depicts two devices connected in series. **c** Voltage and current profiles when the capacitor (470 μF) was hooked up to the digital hydrometer shown in the inset. **d** Voltage profiles for one module (m₁) and two modules connected in parallel (m₁ and m₂), followed by charging a capacitor (470 μF) as a function of time under ΔT = 7.6 °C. Each module consists of 4 devices connected in series.

water proliferates with time, the potential in the hotter side is raised, while that in the colder side is lowered, escalating the potential difference between the two electrodes (Fig. 3g). When the temperature becomes uniform, the water in the colder side is redistributed, and thereby the potential difference converges to zero (Fig. 3h). This working mechanism also explains the peak TtoE factor at 50% RH rather than monotonically increasing trends with a higher RH in the literature. When the water uptake in PANI:PSS is too high, it is hard to induce a significant difference in the water concentrations on the two electrodes, resulting in a lower voltage. Conversely, low water uptake is unfavorable to the thermodiffusion of water due to the limited amount of water[12].

Our device is promising for various applications, including sensors and energy harvesters. For example, the colossal TtoE factor from this study, which is several orders of magnitude larger than those of thermocouples, could give a substantial voltage response to a temperature difference and thereby ameliorate the signal-to-noise ratio. It is worth mentioning that the corrosion of the carbon steel resulted in only ~18-μm reduction for six months

(see Supplementary Fig. 11a). Even if we assume continuous dissolution of carbon steel, 0.4-mm-thick carbon steel could last longer than 10 years. Here we demonstrated its functionality as an energy harvester using the device with the intermediate oxidation layer. A single-unit device was connected to a capacitor (470 μF) and a load resistor (10 kΩ) in parallel with S₁, S₂, and S₃ switches (Fig. 4a). Under a temperature difference of 8 K, the open-circuit voltage reached 360 mV with "on" state only for S₁ switch. Then, a capacitor (470 μF) was charged to 350 mV by closing S₂, and subsequently the capacitor was discharged by a load resistor (S₂ and S₃ were closed). After repeated charge/discharge cycles, voltage approached zero when the temperature of the device became uniform.

The output voltage was further boosted by connecting 2, 4, 6, and 8 units in series (Fig. 4b). Under the temperature difference of 7.6 °C and 3.9 °C, serially joined eight modules produced 2.8 V and 1.45 V, respectively. The linear relationship between voltage generation and the number of modules indicates that the output voltage can be elevated by serial connection. Figure 4c displays that a digital hygrometer with a LCD screen was powered by four

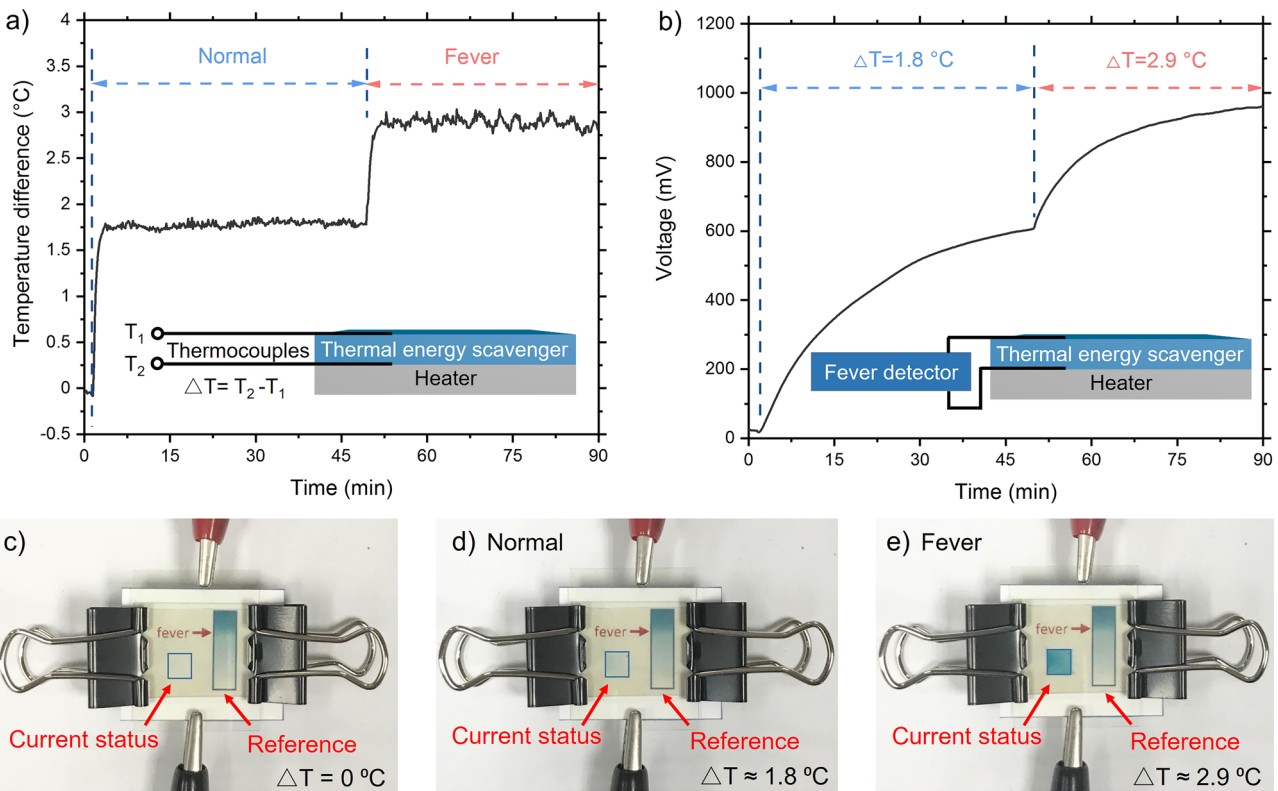

**Fig. 5 Operation of an electrochromic fever detector by a thermo-hydro-electrochemical energy harvester. a** Temperature difference as a function of time when heat is dissipated at the rates of 360 W m$^{-2}$ and 580 W m$^{-2}$, to mimic normal and fever conditions, respectively. The inset illustrates the temperature-measurement configuration with a heater. **b** Voltage profile as a function of time from serially connected four devices under the temperature differences of 1.8 and 2.9 °C. The inset illustrates that the thermal energy harvester was directly connected to the fever detector. **c** A photograph of the fever detector with an electrochromic display window showing the current status and a reference color bar when heat is not supplied to the device ($\triangle$T = 0 °C). **d** When $\triangle$T ≈ 1.8 °C, the color in the status window became light blue, indicating the device was working under a normal condition. **e** When $\triangle$T ≈ 2.9 °C, the status window displayed dark blue above the fever color in the reference bar.

serially coupled modules with a 470-µF capacitor. When the hygrometer started to work, voltage and current were recorded. We also configured both current and voltage with combined parallel and serial circuits. We made a module consisting of serially joined four units, and two modules ($m_1$ and $m_2$) were hooked up in parallel along with a 470-µF capacitor, and the voltage profile of the two modules (closed $S_1$, $S_2$, and $S_3$) was compared with that of one module ($m_1$ only, $S_1$ and $S_3$ were closed) (Fig. 4d). Initially, the open-circuit voltage was identical for both circuits, but the capacitor was charged more rapidly with two modules, showing ~2.5 min for one time constant (63%) compared with ~9 min with one module. This elucidates that our device offers options for optimizing voltage and current for a system of interest.

Based on the performance assessment, we have developed a self-sustainable fever detector, which could be utilized for early detection and continuous monitoring of viral infectious diseases for humans and livestock. Fever detection based on conventional one-time measurement may suffer from false-negative results because it could be strongly affected by human errors, environmental conditions, and stage of infection, in addition to a risk of cross-infection. These drawbacks could be mitigated by continuous monitoring with low-cost and self-sustainable sensors and electrochromic display devices (see Supplementary Fig. 12 and Fig. 5c, d, e). To visualize the temperature changes, we integrated an electrochromic display with serially-connected four devices with the fully-developed oxidation layer. The electrochromic display was made of Prussian white (see Supplementary

Fig. 13a). When electricity was supplied to the display, the color was changed from white to blue, and then, without electricity, from blue to white, reversibly. To mimic a situation with a fever, the heat flux from a human without and with a fever was assumed to be 360 W m$^{-2}$ and 580 W m$^{-2}$, respectively, which have yielded the temperature differences ($\triangle$T) of 1.8 °C and 2.9 °C, respectively, across the device (Fig. 5a) (See Section 8 in SI)[42–45]. Under the temperature differences, voltages of ~0.6 V and ~1 V were observed (Fig. 5b). Before the operation of the device, the status window on the left in Fig. 5c is white. When $\triangle$T was 1.8 °C, the color of the window was slightly changed to light blue, indicating the temperature is below a fever on the reference bar (Fig. 5d). Further increase in $\triangle$T to 2.9 °C resulted in a darker-blue window, indicating fever or higher temperature. For other or more sophisticated applications, the number of devices can be readily altered, and the color of the reference bar can be adjusted for desired temperature ranges.

In summary, we discovered a method of generating a large TtoE factor, −87 mV K$^{-1}$ at 22 °C and 50% RH by utilizing the change in the corrosion potential due to the thermodiffusion of water, through a series of systematic and rigorous experimental studies for unveiling the working mechanisms. We also substantiated the developed thermo–hydro-electrochemical conversion concept by powering electronic devices, including a fever detector that can be distributed to many unspecified people at public places at a low price. We anticipate that this study opens up and facilitates subsequent research for achieving even higher

TtoE factors as well as developing self-sustainable electronic devices, including disposable, low-cost, and compact sensors.

## Methods

**Materials.** Polystyrene sulfonic acid (PSS-H) (M.W. 75000, 30 wt%; Alfa Aesar), aniline (99+ %; Alfa Aesar), hydrochloric acid (HCl, 36.5–38.0%, ACS; Macron Fine Chemicals), carbon steel shim (1008–1010 carbon steel, thickness: 0.005 inch; Precision Brand Products, Inc.), graphite foil (≥99.8% metals basis, thickness 0.254 mm; Alfa Aesar), iron(III) chloride (anhydrous; Sigma Aldrich), iron(II) chloride (tetrahydrate, 98%; Alfa Aesar), iron foil (iron ≥99.99% metal basis, thickness 0.1 mm; Alfa Aesar), ammonium persulfate (ACS; J.T. Baker), deionized (DI) water (>18 MΩ cm$^{-1}$), potassium ferricyanide (K$_3$Fe(CN)$_6$, 98.5%; Acros Organics), tetraethylammonium perchlorate (TEAP, 98%; Alfar Aesar), ITO glass (100 Ω sq$^{-1}$; Nanocs), Nafion 115 membrane (Fuel Cell Earth), and hydrazine (anhydrous, 98%; Sigma Aldrich). The digital hygrometer with LCD screen was purchased from Linkstyle.

**Material synthesis.** The following procedure was used to synthesize the poly-electrolyte made of polyaniline and polystyrene sulfonate (PANI:PSS)[30]. First, 30-wt% PSS-H (30 g) and aniline (4 mL) were added to DI water (80 mL), and the solution was stirred for 1 h. Ammonium persulfate solution was diluted in DI water (50 mg of ammonium persulfate per mL of water). Then, 40 mL of the aqueous ammonium persulfate solution was slowly dropped into the PANI:PSS solution for 30 min using a syringe pump while the solution was stirred using a magnetic bar. The solution was stirred for 5 min and then stored overnight. Subsequently, the solution was poured into acetone (1 L) for precipitation, and then decanted the solution. The collected precipitate was washed by acetone five times, and then dried at 50 °C in an oven to obtain dark-green PANI:PSS (~13 g). The dried PANI:PSS precipitate dissolved in DI water (20 wt%) by stirring for 2 h, and then the hydrochloric acid was added to the PANI:PSS solution (10 vol% of the hydrochloric acid in the PANI:PSS solution). After the solution was stirred for 1 h, the solution (~3 g) was drop-casted on two carbon steel foils (dimension is 2 cm by 2 cm), and two pieces were put together before they were fully dried (typically after 12 h in a fume hood). The sample was left in a fume hood for five days at room temperature. During the drying process, carbon paint was coated on the outer side of the carbon steel for good electrical connection with lead metal tab for electrical measurements. The thickness of the device was ~2 mm.

**TtoE factor and electrochemical impedance spectroscopy (EIS) measurements.** TtoE factor was measured using our custom-built setup (see Supplementary Fig. 3) in a humidity-controlled chamber. Peltier devices with aluminum block were used for controlling temperature. The device was placed in the middle of two Peltier devices to control the temperature difference. Two thermocouples were placed on the top and bottom of the device to measure the temperature, while two copper wires were used to measure the voltage between the two electrodes. The temperature difference was varied by altering the current passing through the Peltier device. We took data after the sample was left in one particular RH level for at least 8 h to ensure the sample reached the steady state. Voltage as a function of time was measured at various temperature differences, typically 6–8 points, and the linear slope in temperature difference (x axis) vs. voltage (y axis) was sought for finding the TtoE factor (flipped the sign like conventional thermopower). For EIS measurements, our device was kept under different humidity levels for 8 h, and then it was scanned over a frequency range between 0.1 and 10$^6$ Hz.

## Data availability

The data that support the findings of this study are available from the corresponding author upon reasonable request.

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

## Acknowledgements

The authors acknowledge financial support from US National Science Foundation (CBET 1805963).

## Author contributions

C.Y. and Y.Z. conceived the idea, carried out the experiments and analyses, and wrote the paper. A.S. and A.C. assisted the experiments.

## Competing interests

The authors declare no competing interests.
