## [Peer Review File · Nature Communications]

Colossal Thermo-Hydro-Electrochemical Voltage Generation for Self-sustainable Operation of ElectronicsPeer Review File

Reviewer comments, first round –

Reviewer #1 (Remarks to the Author):

The manuscript by Zhang et al. achieved an extremely large thermopower with carbon steel electrodes and PANI:PSS as a solid-state polyelectrolyte. The authors did thorough studies to unveiling that the high thermopower is originated from the change in the corrosion potential due to the thermo-diffusion of water. The thermal energy harvester exhibits promising and attractive applications. However, some discussion of the results is not clear and accurate. We don't think it's suitable for publishing on Nature Communications.

- 1. There are serious contradictions in the author's explanation of the source of the huge potential difference. The author suggests that "Water is an electrolyte in corrosion reactions so a reduction in water renders the corrosion overpotential higher." In my understanding, the potential of the black line should be changed, rather than the green line in Fig. 3d. In addition, according to the relation (1 and 2) given by the author, the potential difference comes from the resistance difference caused by water content.*
- 2. Following the first point, how can the current at the cold and hot end be different? The author needs to further explain the meaning of Fig. 3d.*
- 3. Why the Seebeck coefficient is negative, while it is protons in PANI:PSS migrate from the hotter side to the colder side in the device with graphite electrode?*
- 4. The author should give the variation of the potential of single side electrode with temperature and humidity humidity, to exclude some interference.*
- 5. The change of corrosion potential due to the thermo-diffusion of water could explain the positive slope in temperature difference vs. voltage. But it is not clear why it results in such a large thermopower here. If it is caused by the change of water concentration as the author said, it's could be estimated easily.*
- 6. The authors attributed the voltage generation drop under high humidity to the fact that it is hard to induce a large difference in the water concentrations on the two electrodes. However, in Fig 3b, the voltage generation can reach even 400 mV under different RH, indicating a significant water concentration difference can be achieved under high RH.*
- 7. The labeled potential in the inset diagram of Fig 3b might be wrong.*
- 8. The "(a)" in the capture of Figure 2a are missed.*

Reviewer #2 (Remarks to the Author):

This is a well written manuscript, supported by a nice series of experimental data. Actually, I wish I could be more supportive. But there are a number of fundamental points, listed below, that prevents me to support this manuscript for publication in Nat Comm.

- Motivations: measuring temperature is not really a problem in combatting Covid. A battery-less thermometer would give minimal benefits in my modest opinion. What is the problem in changing the battery every 6-12 months or so?

- *Underlying mechanism: This is clearly not a thermoelectric Seebeck effect, as also demonstrated by the authors themselves. Terms like “colossal thermopower” are then just misleading.*
- *Practicality: As shown in Fig. 2 and Fig.4, the voltages stabilise over very long period of time (tens to hundreds of minutes!), as expected from ion diffusion as opposed to electron diffusion. This not only casts doubts on the accuracy of the measured “thermopower” but also highlights a huge limitation of this phenomenon: the extremely slow response time.*

Reviewer #3 (Remarks to the Author):

The manuscript reports a large thermopower that arises through a change in the corrosion potential of carbon steel electrodes based on thermally driven water diffusion. The findings of such a large thermopower are interesting and the authors do a decent job of determining the origins for such a large thermopower. Parts of the discussion are confusing and it appears that some of this confusion stems from the authors reversal of the typical sign convention for the voltage (thermopower). The fact that the large thermopower results from electrochemical reactions promoted by increasing water concentration makes this device different than a traditional thermoelectric material. It is not clear whether these reactions are reversible, but it does not appear so, which makes their device different than a traditional thermogalvanic cell. The manuscript is likely to be of high interest, but the manuscript needs to be more clearly written. The following are my comments on the manuscript:

A positive thermopower, by convention, means that the colder electrode is more positive than the hotter electrode. This sign convention appears to be reversed in the paper, which made it difficult to read through.

The inset in Figure S5b is confusing. In general, labeling the electrode terminal as positive and negative, even though they are not necessarily positive or negative, as is done throughout the paper and SI figures is confusing. In Figure S5b the polyelectrolyte layer shows positive charges accumulating at the cold side – which would appear to make the voltage positive and not negative. The authors assign the thermovoltage as negative though. The description in the paper is also incorrect, “When protons in PANI:PSS migrate from the hotter side to the colder side, a negative slope is obtained.” The thermally driven diffusion of protons from the hotter side to the colder side should result in a positive Seebeck coefficient (positive slope to the voltage vs. temp plot).

The authors mention that water movement from the hotter side to the colder side results in an increase in potential of the hotter side. If the potential of the hotter side increases with temperature due to water diffusion away from that electrode, then the thermopower should be negative. Figure 2a shows a positive Seebeck coefficient. Again, I think that much of the confusion when reading this manuscript stemmed from improper sign convention. If the proper sign convention is indeed being used, then the discussion throughout most of the paper does not make sense.

The focus on fever detection in the manuscript should be reduced. Since the thermoelectric device is only sensitive to a temperature difference, using for fever detection would be extremely difficult. For example, with colder ambient conditions it would look like everyone has a fever and under warmer ambient conditions it would appear that no one did. Now, the whole opening paragraph is about fever detection and much of the abstract is too. I recommend focusing more on the actual science

and engineering reported.

What other elements are in the carbon steel electrodes? Is it possible that these are also contributing to the thermopower?

REFEREE 1

Reviewer's comments

The manuscript by Zhang et al. achieved an extremely large thermopower with carbon steel electrodes and PANI:PSS as a solid-state polyelectrolyte. The authors did thorough studies to unveiling that the high thermopower is originated from the change in the corrosion potential due to the thermo-diffusion of water. The thermal energy harvester exhibits promising and attractive applications. However, some discussion of the results is not clear and accurate. We don't think it's suitable for publishing on Nature Communications.

1. There are serious contradictions in the author's explanation of the source of the huge potential difference. The author suggests that "Water is an electrolyte in corrosion reactions so a reduction in water renders the corrosion overpotential higher." In my understanding, the potential of the black line should be changed, rather than the green line in Fig. 3d. In addition, according to the relation (1 and 2) given by the author, the potential difference comes from the resistance difference caused by water content.

Authors' responses

We truly appreciate your valuable time and efforts in reviewing this manuscript.

The anodic and cathodic reactions in Fig. 3d are:

The black line in Fig. 3d indicates the Tafel curve of the cathodic reaction where water is the product of this reaction, not a reactant. The water in this system is an electrolyte to form a corrosion cell on the carbon steel surface, which affects the anodic reaction. When Fe is oxidized to become Fe^{2+} , water is necessary as an electrolyte. Therefore the amount of water governs the anodic reaction, not the cathodic reaction.(1) The potential difference comes from the resistance difference caused by the water content, which is called the overpotential difference in our manuscript. The green line should be changed in the Evans diagram, and thus there are no contradictions in our description and Fig. 3d.

Reviewer's comments

2. Following the first point, how can the current at the cold and hot end be different? The author needs to further explain the meaning of Fig. 3d.

Authors' responses

Based on the first and second comments, we think it is better to add more texts regarding the cathodic and anodic reactions to avoid misunderstanding, if any. Our system is not the same as batteries, which typically consist of a cathode and an anode, referring to two different electrodes. The cathodic and anodic reactions in our case occur in one electrode (carbon steel), as illustrated in Fig. S11b. There is corrosion current in one electrode between cathodic and anodic sites (see the red arrow).

Fig. S11. (b) Illustration of the carbon steel electrode where FeOOH covered its surface with water in between.

Suppose that the water migration makes the water content in one electrode is higher than the other. Then a potential difference between the two electrodes is created, as illustrated in Figure R1. Perhaps it is easier to understand if one recalls corrosion is facilitated in a humid environment.

Figure R1. A potential difference created between a hotter electrode and a colder electrode.

Reviewer's comments

3. Why the Seebeck coefficient is negative, while it is protons in PANI:PSS migrate from the hotter side to the colder side in the device with graphite electrode?

Authors' responses

The device with graphite electrodes is described in Fig. S5 (Supporting Information 6.1). We are sorry for the confusion. In Fig. S5, we used “slope” (dV/dT) rather than thermopower, and the slope is negative. On the other hand, the “slope” in the inset of Fig. 2a is positive. They have the opposite signs. According to the typical sign convention of thermopower (S), the minus sign needs to be placed in front of the slope ($S = -dV/dT$). Therefore, a negative slope means a positive thermopower, and a positive slope means a negative thermopower. In our manuscript, we intentionally used “slope” instead of “thermopower” (or Seebeck coefficient) to explain the transport mechanisms because we found that the negative sign in thermopower often made readers confused. In Figure 2b, the plot shows the “absolute” values of thermopower to compare thermopower values in the literature (i.e., mixed positive and negative values). The thermopower values from our devices made of carbon steel are negative. In the manuscript texts, we wrote absolute values, which were mentioned in the figure caption, but we should have clearly mentioned this in the main texts. We believe this is the origin of the confusion. Therefore, in the revised manuscript, we inserted the negative sign for our thermopower values except Figure 2b because it is hard to compare positive and negative values together in one plot.

Reviewer's comments

4. The author should give the variation of the potential of single side electrode with temperature and humidity humidity, to exclude some interference.

Authors' responses

Figure 3b shows the variations of the water content under different humidity levels. It is difficult to get useful conclusions by changing both temperature and humidity simultaneously because the humidity and water content are coupled. Perhaps the reviewer may want to see the influence of temperature without altering humidity (or water content), and vice versa. To isolate the temperature effect from humidity changes in a single electrode, we designed an experiment similar to Fig. 3a and 3b, and then measured the potential change of a single side electrode at different temperatures. One carbon steel electrode was taken out from a device, and left under different temperatures for 12 hours at 50 %RH. Then, we assembled the device and measured the voltage (the positive pole was replaced with the electrode under different temperatures). The environment temperature and RH were maintained to be 22 °C and 50%, respectively, during the experiment.

As shown in Fig. R2, the slope in the temperature-voltage profile is only ~ 1.15 mV/K (thermopower of -1.15 mV/K), which is much smaller than those of our devices in Fig. 2b. The voltage variations under the humidity change without any temperature difference are an order of magnitude higher, as displayed in Fig. 3b.

Figure R2. Voltage measurement results when one electrode was left under different temperatures for 12 hours at 50 %RH.

Reviewer's comments

5. The change of corrosion potential due to the thermo-diffusion of water could explain the positive slope in temperature difference vs. voltage. But it is not clear why it results in such a large thermopower here. If it is caused by the change of water concentration as the author said, it's could be estimated easily.

Authors' responses

We have verified the working mechanisms with multiple independent experiments, as described in Fig. 3 and Fig. S5-S11. In brief, Fig. 3c shows that the intensity of OH bonding in ATR-FTIR became stronger on the colder side, which means that the water concentration became higher on the colder side. The change in the water content of the electrode causes large potential changes (Fig. 3a and 3b). Figure R2 shows the sole temperature effect without altering the water content, which confirms the temperature difference by itself does not provide a large thermopower. The change in the water content is not easily noticed. Figure S11c indicates that the change in the actual amount of water is only a few mg. The mass of the electrode is about 500 mg, and the mass of the whole device is about 1.3 g. This change in the water mass cannot be easily noticed unless careful

experiments are carried out. It should be noted that Fig. 3b shows “change in water mass”, not the mass of the electrode or device.

Figure S11c. Mass change in a single carbon steel electrode after the sample was left under different relative humidity for 12 h.

Reviewer’s comments

6. The authors attributed the voltage generation drop under high humidity to the fact that it is hard to induce a large difference in the water concentrations on the two electrodes. However, in Fig 3b, the voltage generation can reach even 400 mV under different RH, indicating a significant water concentration difference can be achieved under high RH.

Authors’ responses

In Fig. 3b, one electrode was detached from the device, and the water content of the electrode can be independently changed by exposing it to different RH environments. In actual device, the total water content is maintained, meaning that increasing the water content of one electrode decreases that of the other electrode. Therefore, more water in the device doesn't mean that a larger water concentration difference between the two sides can be achieved. Here “difference” is important. We can image our device with a bottle of water, as illustrated in Fig. R3. When we compare the amount of water contained in the top half and bottom half, the large “difference” can be obtained from Case B₂, not B₃ (which has the large amount of water).

Figure R3. A bottle is filled with three different amounts of water. Suppose we are comparing the amount of water filled in the top half and bottom half. The difference in the amount of water occupying the top half and bottom half is the largest when the bottle is filled a half (Case B₂) because the top half is empty and the bottom half is completely filled. Although Case B₃ has the largest amount of water, the difference is not the largest.

Reviewer’s comments

7. The labeled potential in the inset diagram of Fig 3b might be wrong.

Authors’ responses

The labels for the potential in Fig. 3b are correct. The left-hand side is $V > 0$. This means that the top electrode with less water is more positive, and vice versa.

Reviewer’s comments

8. The “(a)” in the capture of Figure 2a are missed.

Authors’ responses

Thank you for the comment. We added “(a)” to the revised manuscript.

REFEREE 2

Reviewer's comments

This is a well written manuscript, supported by a nice series of experimental data. Actually, I wish I could be more supportive. But there are a number of fundamental points, listed below, that prevents me to support this manuscript for publication in Nat Comm.

- Motivations: measuring temperature is not really a problem in combatting Covid. A battery-less thermometer would give minimal benefits in my modest opinion. What is the problem in changing the battery every 6-12 months or so?

Authors' responses

We truly appreciate your valuable time and efforts in reviewing this manuscript.

The temperature measurement is not a remedy for COVID, but it is one of the practical approaches in alleviating the spread of the virus. It is necessary to use multiple approaches together because there is no single solution in overcoming COVID at this point. Most experts mention that the deadly COVID and mutants are likely to stay with us for many years. Fever is the most common symptom, so one-time temperature measurements are often carried out before passengers take airplanes or people enter buildings. However, the one-time measurement is not very effective because it is easily affected by temporary changes in body temperatures (e.g., cold weather, etc.). We agree that a battery is an excellent power source, but it may not be an ideal choice for some applications, particularly low-cost and disposable applications. Suppose that the temperature monitoring device is distributed to all the customers entering a mall every day. Then the cost would be a big concern when a battery-powered, disposable device is used. It is not desirable to share such devices due to the risk of infection, so they should be disposable. Our device is made of inexpensive materials, which will be much cheaper than battery-powered devices. Ours can be configured to distribute to many unspecified people at public places at an extremely low price. Farmers could also use cheap and disposable patches on livestock to detect some diseases such as Swine flu and others. The electric chromic display can be directly used with our energy scavenging device, but batteries require additional switching devices to make the current supply on/off depending on temperature.

Reviewer's comments

- Underlying mechanism: This is clearly not a thermoelectric Seebeck effect, as also demonstrated by the authors themselves. Terms like “colossal thermopower” are then just misleading.

Authors' responses

Yes, this is not a Seebeck effect. We used the term, “thermopower” for the thermally-induced voltage per unit temperature difference, and intentionally avoided using the Seebeck coefficient or Seebeck effect. Thermopower in the unit of V/K has been used not only for the Seebeck effect but also for the Soret effect (thermo-diffusion of ions) (2-4), thermogalvanic cell (5-7), etc. In fact, we are eager to create a new word for this, but could not find a suitable word that readers undoubtedly accept for the voltage generated by the thermo-diffusion of water due to a temperature difference. A large portion of this manuscript has been devoted to validate and explain the working principle, stating that this is different from the conventional Seebeck effect. It is hard to miss this point in this manuscript.

Reviewer's comments

- Practicality: As shown in Fig. 2 and Fig.4, the voltages stabilise over very long period of time (tens to hundreds of minutes!), as expected from ion diffusion as opposed to electron diffusion. This not only casts doubts on the accuracy of the measured “thermopower” but also highlights a huge limitation of this phenomenon: the extremely slow response time.

Authors' responses

We agree that it is premature to say something about practicality. Considering this is the first paper, it is too early to affirm that it is not practical. It takes time to reach the saturation point, but it is not too bad to get to the point of one time constant (63% of its saturated value). When the capacitor (470 μ F) was charged with two modules, it took \sim 2.5 min for one time constant (Fig. 4d). It is not necessary to wait until it reaches the saturation point. Thermal responses are not as fast as electronic counterparts. Our device is suitable for applications that do not necessitate fast responses. For example, fever detection at every minute is not required. We target niche applications when power grids are not accessible; battery replacement is not easy; and the cost is an important consideration. Considering that this is the first report, there will be ample opportunities for further improvements, such as the response time. To reduce the time for water transport, we are currently working on fabricating cellulose-based straight nano-channels made out of wood by following the work of Li et al. (Nat. Mater. 18, 608-613 (2019)). According to the reference paper, fast ion transport can be achieved with straight and nano-pores made of cellulose.

REFEREE 3

Reviewer's comments

The manuscript reports a large thermopower that arises through a change in the corrosion potential of carbon steel electrodes based on thermally driven water diffusion. The findings of such a large thermopower are interesting and the authors do a decent job of determining the origins for such a large thermopower. Parts of the discussion are confusing and it appears that some of this confusion stems from the authors reversal of the typical sign convention for the voltage (thermopower). The fact that the large thermopower results from electrochemical reactions promoted by increasing water concentration makes this device different than a traditional thermoelectric material. It is not clear whether these reactions are reversible, but it does not appear so, which makes their device different than a traditional thermogalvanic cell. The manuscript is likely to be of high interest, but the manuscript needs to be more clearly written. The following are my comments on the manuscript:

A positive thermopower, by convention, means that the colder electrode is more positive than the hotter electrode. This sign convention appears to be reversed in the paper, which made it difficult to read through.

Authors' responses

We truly appreciate your valuable time and efforts in reviewing this manuscript.

In our manuscript, we used the absolute value of thermopower. We noticed that this would have caused the troubles. We should have made it clear in the main text, albeit indicated in the figure caption. For the comparative study between graphite and carbon steel devices, we used “slope (dV/dT)” to describe voltage per kelvin. In the revised manuscript, we inserted the negative sign for the thermopower values of our devices by following the conventional sign of thermopower (rather than the absolute value) except Figure 2b because it is hard to compare positive and negative values together in one plot.

Reviewer's comments

The inset in Figure S5b is confusing. In general, labeling the electrode terminal as positive and negative, even though they are not necessarily positive or negative, as is done throughout the paper and SI figures is confusing. In Figure S5b the polyelectrolyte layer shows positive charges accumulating at the cold side – which would appear to make the voltage positive and not negative. The authors assign the thermovoltage as negative though. The description in the paper is also

incorrect, “When protons in PANI:PSS migrate from the hotter side to the colder side, a negative slope is obtained.” The thermally driven diffusion of protons from the hotter side to the colder side should result in a positive Seebeck coefficient (positive slope to the voltage vs. temp plot).

Authors’ responses

We have indicated the polarity of the terminals, indicating how the terminals were connected to the multimeter. Without this description, it is hard to tell if the voltage is positive or negative. The description in our manuscript is a negative “slope” (not thermopower). In our manuscript, we intentionally used “slope” instead of “thermopower” (or Seebeck coefficient) to explain the transport mechanisms because we found that the negative sign in thermopower often made readers confused. We use this negative slope to compare with the positive “slope” in the carbon steel device. Again, the absolute value of thermopower would have caused this confusion. Therefore, we used the typical sign convention of thermopower in the revised version of our manuscript. We also added figure captions stating that “the slope is negative, so thermopower is positive” in Fig. S5.

Reviewer’s comments

The authors mention that water movement from the hotter side to the colder side results in an increase in potential of the hotter side. If the potential of the hotter side increases with temperature due to water diffusion away from that electrode, then the thermopower should be negative. Figure 2a shows a positive Seebeck coefficient. Again, I think that much of the confusion when reading this manuscript stemmed from improper sign convention. If the proper sign convention is indeed being used, then the discussion throughout most of the paper does not make sense.

Authors’ responses

Figure 2a displays experimental data, showing a positive “slope” (dV/dT). According to the typical sign convention of thermopower (S), the minus sign needs to be placed in front of the slope ($S = -dV/dT$). Therefore, all the thermopower values are negative. We are sorry for the confusion. We have modified all the signs in the revised manuscript.

Reviewer’s comments

The focus on fever detection in the manuscript should be reduced. Since the thermoelectric device is only sensitive to a temperature difference, using for fever detection would be extremely difficult. For example, with colder ambient conditions it would look like everyone has a fever and under warmer ambient conditions it would appear that no one did. Now, the whole opening paragraph is about fever detection and much of the abstract is too. I recommend focusing more on the actual

science and engineering reported.

Authors' responses

Thank you for your suggestion. We have changed the title and reduced texts regarding fever detection in the revised manuscript.

Reviewer's comments

What other elements are in the carbon steel electrodes? Is it possible that these are also contributing to the thermopower?

Authors' responses

The main element in the carbon steel is iron (>99%). Various other elements with small fractions are included in the carbon steel. The other elements might affect the corrosion rates, but their concentrations are not high enough to form something significant, unlike Fe (e.g., FeOOH).

References:

- 1 Gardiner, C. & Melchers, R. Corrosion of mild steel in porous media. *Corros Sci* **44**, 2459-2478 (2002).
- 2 Cheng, H., He, X., Fan, Z. & Ouyang, J. Flexible quasi-solid state ionogels with remarkable Seebeck coefficient and high thermoelectric properties. *Adv. Energy Mater.* **9**, 1901085 (2019).
- 3 Kim, B., Hwang, J. U. & Kim, E. Chloride transport in conductive polymer films for an n-type thermoelectric platform. *Energy Environ. Sci.* **13**, 859-867 (2020).
- 4 Li, T. *et al.* Cellulose ionic conductors with high differential thermal voltage for low-grade heat harvesting. *Nat. Mater.* **18**, 608-613 (2019).
- 5 Duan, J. *et al.* Aqueous thermogalvanic cells with a high Seebeck coefficient for low-grade heat harvest. *Nat. Commun.* **9**, 5146 (2018).
- 6 Kim, T. *et al.* High thermopower of ferri/ferrocyanide redox couple in organic-water solutions. *Nano Energy* **31**, 160-167 (2017).
- 7 Han, C.-G. *et al.* Giant thermopower of ionic gelatin near room temperature. *Science* **368**, 1091-1098 (2020).

Colossal Thermopower for Battery-less and Self-sustainable Operation of
Electronics

Deleted: ,

Deleted: Fever Detection Enabled by Colossal Thermopower

Yufan Zhang¹, Ahrum Sohn², Anirban Chakraborty², Choongho Yu^{1,2} *

¹ Department of Materials Science and Engineering, Texas A&M University, College Station, Texas, 77843 USA

² Department of Mechanical Engineering, Texas A&M University, College Station, Texas, 77843 USA

*Corresponding author: chyu@tamu.edu

Keywords

Thermopower; Thermoelectric; Corrosion; Energy harvesting; Thermo-diffusion

Deleted: ; Pandemic

Abstract

Thermoelectrics are suited to converting dissipated heat into electricity for operating electronics, but the small voltage ($\sim 0.1 \text{ mV K}^{-1}$) from the Seebeck effect has been problematic. Here a new approach can generate the largest ever reported thermopower, -87 mV K^{-1} in a typical ambient condition with two components, low-cost carbon steel electrodes and a solid-state polyelectrolyte made of polyaniline and polystyrene sulfonate (PANI:PSS). We discovered that the thermo-diffusion of water in PANI:PSS under a temperature gradient induced less (or more) water on the hotter (or colder) side, raising (or lowering) the corrosion overpotential in the hotter (or colder) side and thereby generating output power between the electrodes. The practicality of the colossal thermopower has been validated by developing a fever indicator with only dissipated thermal energy, showing color changes with and without fever reversibly. Our new findings are expected to facilitate subsequent research for further increasing thermopower and utilizing dissipated thermal energy.

Deleted: Unprecedented pandemic has been caused by rapidly spreading and deadly infectious disease that commonly accompanies fever. For early detection and monitoring, self-powered fever sensors without batteries could offer an inexpensive, disposable, and compact solution.

Deleted: energy scavenging

Deleted: is particularly promising in

Deleted: utilizing

Deleted: magnitude of

Deleted: /

Deleted: generated

Deleted: traditional electronic

Deleted: is too small

Deleted: only on the order of $0.01\text{--}0.1 \text{ mV/K}$

Deleted: we present

Formatted: Superscript

Deleted: to

Deleted: highest

Deleted: /

Formatted: Superscript

Deleted: ($22 \text{ }^\circ\text{C}$ and 50% relative humidity)

Deleted: only

Deleted: With thorough studies unveiling the working principle, w

Deleted: voltage and

Deleted: and operating

Deleted: systematic study is

Deleted: as well as

Introduction

Thermoelectricity refers to converting heat to electricity or vice versa, and has been used for various applications including thermocouples and Peltier devices. It is desired to generate a large voltage per temperature difference (*i.e.*, large thermopower or Seebeck coefficient in the unit of V K^{-1}), resulting from the thermo-diffusion of electrons called the Seebeck effect¹. As for traditional solid-state inorganic and recent organic thermoelectric materials, small thermopower values on the order of 0.01~0.1 mV K^{-1} near room temperature have been one of the major hurdles in acquiring high device performances²⁻⁷. In the last several years, the thermo-diffusion of ions, called the Soret effect, has been utilized to aggrandize thermopower to 8 mV K^{-1} with polystyrene sulfonic acid due to the substantial difference in thermo-diffusion between mobile protons and immobile anions⁸. While it has been difficult to have greater than several mV K^{-1} , several recent studies yielded gigantic thermopower⁹⁻¹² including 18 mV K^{-1} from the migration of chloride ions in n-type mixed ionic–electronic conductive polymer composite films¹³, 17 mV K^{-1} partially from KCl in a quasi-solid-state ionic thermoelectric material¹⁴, and 24 mV K^{-1} with sodium ions in cellulose ionic conductors¹⁵. It is worth noting that the thermally-induced voltage is different from moisture-powered generators (MPG) based on ions movement carried by water-diffusion and ion accumulation¹⁶⁻²². MPG works only when environmental humidity changes or water droplets are applied to the device, and it does not respond to a temperature difference.

To induce a thermally-induced voltage on the order of 1~10 mV K^{-1} , a few different mechanisms have been recently reported, including the thermo-diffusion of electron/ion mixture (both Soret and Seebeck effects)^{23,24} and temperature-dependent redox reactions with redox

Formatted: Font: Bold

Formatted: Indent: First line: 0"

Deleted: COVID has paralyzed most of our socio-economic activities in most parts of the world.^{1,2} To prevent a wide spread of viral infection, we need to develop multiple different approaches for "early and fast" detection in addition to conventional time-consuming virus testing.³ Without vaccine and medicine, early and fast detection is crucial. As the most common symptom of viral infection leading to pandemic such as COVID, SARS, MERS, and swine flu is fever, temperature monitoring is possibly a promising and practical measure to identify potential infectees. For example, the majority of COVID patients are known to have had fever. It is common to measure body temperature with an infrared (IR) camera or thermometer when people enter a building, classroom, airplane, etc. However, these one-time measurements often suffer from false negative predictions because the measurements are strongly affected by human errors, environmental conditions, and stage of infection in addition to a risk of cross infection. The aforementioned problems could be greatly mitigated by continuous monitoring of body temperature with compact, inexpensive, disposable, and wearable temperature monitoring device. ¶

Deleted: One of the hurdles for developing such devices is battery, and thermoelectric energy conversion could be an alternative option for power delivery.

Deleted: /K

Deleted: /K

Deleted: /K

Deleted: /K

Deleted: /K

Deleted: /K

Deleted: /K

Deleted: /K

couples in liquid states²⁵⁻²⁷. It appears that solid-state polyelectrolytes utilizing the Soret effect are the best option for high thermopower so far, and their highest thermopower values were obtained at unusually high (70-100%) relative humidity (RH) rather than typical room humidity (~50% RH). Water is an electrolyte for the ions, improving their mobility, and water makes mobile ions readily dissociated from their counter ions^{8,11,23}. However, high water uptake in solid-state polyelectrolytes often causes stability problems due to irreversible water evaporation and swelling. Considering high thermopower is the key to the performance of thermoelectric applications, it is valuable to seek other routes for attaining even bigger thermopower. For example, to acquire a working voltage (> 1 V) of typical wearable electronics with traditional inorganic materials, at least 1000 thermoelectric legs should be serially connected under a temperature difference of 10 °C.

Here we report a new approach based on the variation of potentials caused by a temperature difference. We used readily available carbon steel as electrodes to obtain the colossal thermopower of -87 mV K^{-1} under a typical ambient condition (50% RH, 22 °C). Porous hydrophilic layers were formed on the carbon steel, and a hygroscopic solid-state polyelectrolyte layer was used between the two electrodes. We discovered that, upon imposing a temperature difference, the thermo-diffusion of water from the hotter side to the colder side altered the water uptake in the electrode, differentiating the potential of the two electrodes. Based on the new mechanism, a battery-less, self-sustainable fever detection device has been operated as a proof of principle, which could be helpful in the early and fast detection of fever commonly observed from a viral infection such as COVID, SARS, MERS, and swine flu pandemic. The following include fabrication and characterization of materials and devices, investigation of working principles, and applications.

Deleted: -

Deleted: /K

Deleted: useful for

Results and discussion

Formatted: Font: Bold, Font color: Auto

Our device consists of polyaniline and polystyrene sulfonate (PANI:PSS) as a solid-state electrolyte and carbon steel foils as electrodes (Fig. 1a). PANI:PSS powders (Fig. S1 for FTIR) were synthesized with polystyrene sulfonic acid (PSS-H) and aniline²⁸, and then they were dissolved in deionized water with hydrochloric acid. The solution was drop-casted on two carbon steel electrodes, and two pieces were assembled before they were fully dried. After the assembly, the sample was left in a fume hood. During this time period, the surface of the carbon steel was corroded, forming a new layer between PANI:PSS and electrodes, as shown in Fig. 1b. We observed a porous layer composed of few micron long nanorods (Fig. 1c) under a flat PANI:PSS layer (Fig. 1d). The XRD patterns (Fig. 1e) indicate that the nanorod is made of β -FeOOH²⁹. As PANI:PSS is a hygroscopic material^{8,30}, the water uptake is a strong function of RH in the environment. The amount of water soaked in PANI:PSS under different RH was characterized as a function of time (Fig. S2), and steady-state values are summarized in Fig. 1f. The water uptake in the sample was augmented with higher RH, and was found to be ~15 wt% in PANI:PSS under a typical room environment (50% RH). Transport property measurements were carried out after the water uptake reached steady states.

For thermally-induced voltage measurements, the temperature difference between two electrodes was varied up to ± 6 K, and voltage was recorded as a function of time. Figure 2a shows the generated voltage of the devices with 15% water uptake (RH=50%), and the saturated voltage as a function of temperature difference was plotted in the inset of Fig. 2a. Those of all the other samples are shown in Fig. S3, Table S1, and Table S2. The slopes from the linear fitting are the

absolute values of thermopower, which are plotted against RH along with those from literature (Fig. 2b)^{8,9,11-14,23,31-33}. It should be noted that the sign of the number for the slope should be reversed to get thermopower (*i.e.*, a positive slope means a negative thermopower). We found that thermopower gets bigger as we elongated the oxidation time of the carbon steel in the ambient condition, but thermopower did not further increase after ~60 days. We observed consistent thermopower, $-85 \sim -87 \text{ mV K}^{-1}$ after 120 days and 180 days. The two different cases from fully-developed (60 days) and intermediate (14 days) oxidation layers are shown in Fig. 2b. At 30% RH, the thermopower of the fully-developed case was -48 mV K^{-1} and, under 50% RH, it was boosted up to -87 mV K^{-1} , which is the highest value ever reported, to the best of our knowledge. Even for the intermediate case, -47 mV K^{-1} at 50% RH is much higher than those in the literature.

The difference in thermopower for the two cases mainly comes from the impedance change of the oxidation layer. A greater potential difference between electrodes can be developed when the impedance of the oxidation layer was enlarged. According to the electrochemical impedance spectroscopy results (Fig. S4), the impedance of PANI:PSS on the order of 10Ω was significantly raised to values on the order of $\text{k}\Omega$ with the oxidation layers, and larger impedance was observed from the fully-developed cases. It is interesting to see the distinct humidity dependency from our sample where there is an optimum RH while the others show monotonically increasing thermopower with RH. In fact, the optimum performance at 50% RH is ideal because it is close to that of typical indoor environments. On the other hand, this would be an indicator that the working principle of our system is different from the others.

In the literature reporting high thermopower, the thermo-diffusion of ions (*e.g.*, proton) was

Deleted: whose absolute values

Formatted: Font: Italic

Deleted: -

Deleted: -

Deleted: /K

Deleted: -

Deleted: /K

Deleted: -

Deleted: /K

Deleted: -

Deleted: /K

identified to be the main driver^{8,10-15,34,35}. Here, to test the influence of the thermo-diffusion of ions in PANI:PSS on thermopower, we assembled a cell with graphite foils instead of carbon steel (Fig. S5a). We found that the slope in temperature difference vs. voltage plot in Fig. S5b is opposite to that of our device in the inset of Fig. 2a. When protons in PANI:PSS migrate from the hotter side to the colder side, a negative slope (or a positive thermopower) was obtained. Moreover, thermopower from the device with graphite electrodes was found to be only 0.97 mV K^{-1} , which is much smaller than -87 mV K^{-1} from our carbon steel based device. Therefore it is clear that the working principle of our device is different from those in the literature. Instead, the graphite device is similar to MPG¹⁶⁻²² because they both rely on proton migration. To verify the difference between MPG and ours, we carried out experiments exposing moisture instead of heat to one of the electrodes (Fig. S6). PANI:PSS with our carbon steel electrodes generated $\sim 360 \text{ mV}$ under RH difference of 58%, whereas the graphite electrode device generated only $\sim 0.2 \text{ mV}$. We also tested the influence of $\beta\text{-FeOOH/Fe}^{2+}$ redox couple on thermopower by sandwiching $\beta\text{-FeOOH}$ between PANI/PSS and graphite foils. We observed voltage continuously changed despite constant temperature difference with a largest absolute value of $\sim 1.3 \text{ mV}$ under 4 K difference (Fig. S7). The large thermopower can be obtained only with the steel electrodes, suggesting a corrosion process plays a key role in our system.

Then we investigated if corrosion was caused by the thermo-diffusion of protons in a 3-electrode configuration with the FeOOH-coated carbon steel electrode (taken out from the device) as a working electrode (Fig. S8). Voltage sweeping between a Pt counter and reference electrodes resulted in Tafel curves, showing a corrosion potential of $-0.45 \text{ V vs. Ag/AgCl}$ when the current

Deleted: i

Deleted: thermally-induced voltage

Deleted: (absolute value)

Deleted: /K

Deleted: -

Deleted: /K

Deleted: the moisture-powered device

Deleted: the moisture power device

between the working and counter electrodes approached zero. As HCl was gradually added to have pH of 1, 0.8, and 0.6 (initially pH = 1.25), the potential shifted towards positive values with more protons (or a lower pH). This result denotes more protons make the corrosion potential of carbon steel more positive, which is the same as Fig. S5 where negative voltages under $\Delta T > 0$ (protons on the bottom electrode) were observed. This behavior is opposite to the trend of our device in Fig.

2a. A similar experiment (Fig. S9) also exhibited that voltage was shifted more positively with the addition of protons, confirming that the thermo-diffusion of the proton is not the major contributor in our system. Furthermore, our experimental results in Fig. S10 proved that the corrosion potential in the Tafel curves is not a significant function of temperature.

As voltage generation strongly depends on humidity, we carried out experiments directly showing the influence of water uptake on voltage generation. One of the electrodes in the device was taken out of the device and exposed to environmental conditions whose RH was altered from 50% RH to 20% RH and 70% RH for 12 hours (Fig. 3a). The porous layer on the electrode can accommodate water from the humid environment or release water initially present in the 50% RH condition, as indicated by the mass change in drier and wetter conditions (Fig. 3b). When the electrode was re-assembled, the voltage was remarkably altered, showing higher (or lower) potential with less (or more) water. When water moves from the hotter side to the colder side, the hotter side has a higher potential than the colder side, which agrees with the trend (slope) shown in our system (inset of Fig. 2a).

We used in-situ attenuated total reflectance (ATR) Fourier transform infrared spectroscopy (FTIR) to identify the thermo-diffusion of water in PANI:PSS by comparing the intensity of O-H

Deleted:

Deleted: confirms that the thermo-diffusion of protons to the colder side causes the

Deleted: like Fig. S5, which

stretching band for water, which appears over a broad range near 2800~3700 cm⁻¹ with its peak³⁶ at ~3450 cm⁻¹ while one side of PANI:PSS was being heated (Fig. 3c). All the spectra were normalized by the peak intensity at 2914 cm⁻¹ corresponding to C-H stretching of the CH₂ group in PSS³⁷. The absorbance peak near 3450 cm⁻¹ from the colder side was intensified as the temperature difference was enlarged, suggesting water diffused from the hotter to the colder side. It should be noted that the colder side of the sample was in contact with the FTIR apparatus at room temperature to avoid water condensation, and a temperature lower than room temperature was not applied due to the risk of changes in the water content.

Deleted: which was

Deleted: it was not feasible to apply

Deleted: condensation

The Evans diagram in Fig. 3d explains how thermoelectric voltage was developed. Initially, the corrosion potential is located at the intersection between the oxidation and reduction Tafel curves (green and black lines in Fig. 3d, respectively). Under a temperature gradient, the thermo-diffusion of water reduces the amount of water in the hotter side while that in the colder side increases. Water is an electrolyte in corrosion reactions, so a reduction in water renders the corrosion overpotential higher. Then the Tafel curve for oxidation shifts counterclockwise (red line in Fig. 3d), and then a new potential (crossover point) is established due to the following relation.

38

$$E_i^0 = E_{\text{anode}} + I_i R_i \quad (1)$$

E_i^0 and E_{anode} are the potential of corrosion and anodic reaction, respectively. $I_i R_i$ is the overpotential. I_i is the corrosion current and R_i is the resistance of the electrolyte, where the index i is either the hotter (h) or colder side (c). On the other hand, higher water uptake decreases the overpotential, lowering the crossover point (blue line in Fig. 3d). The newly established two

crossover points between the reduction line and the raised/lowered oxidation lines for the hotter/colder sides create a potential difference between the two electrodes, as follows.

$$\Delta E = E_h^0 - E_c^0 = I_h R_h - I_c R_c \quad (2)$$

The potential difference is functions of the corrosion current and resistance, which are strongly affected by the amount of water in the electrodes ³⁹.

Figure 3e-f illustrate water diffusion and corresponding electrode potentials. Under a uniform temperature, water is homogeneously distributed in PANI:PSS, and the potentials of the top and bottom electrodes are identical (Fig. 3e). When a temperature gradient is created, the water molecules in the hotter side diffuse to the colder side (Fig. 3f). As the amount of migrated water proliferates with time, the potential in the hotter side is raised while that in the colder side is lowered, escalating the potential difference between the two electrodes (Fig. 3g). When the temperature becomes uniform, the water in the colder side is re-distributed, and thereby the potential difference converges to zero (Fig. 3h). This working mechanism also explains the peak thermopower at 50% RH rather than monotonically augmented thermopower with a higher RH in the literature. When the water uptake in PANI:PSS is too high, it is hard to induce a significant difference in the water concentrations on the two electrodes, resulting in a lower voltage. Conversely, low water uptake is unfavorable to the thermo-diffusion of water due to the limited amount of water ¹¹.

Our device is promising for various applications including sensors and energy harvesters. For example, the colossal thermopower from this study, which is several orders of magnitude larger than the thermopower of thermocouples, could give a substantial voltage response to a temperature

difference and thereby ameliorate the signal-to-noise ratio. It is worth mentioning that the corrosion of the carbon steel resulted in only $\sim 18 \mu\text{m}$ reduction for 6 months (Fig. S11a). Even if we assume continuous dissolution of carbon steel, 0.4-mm thick carbon steel could last longer than 10 years. Here we first demonstrated its functionality as an energy harvester using the device with the intermediate oxidation layer. A single unit device was connected to a capacitor ($470 \mu\text{F}$) and a load resistor ($10 \text{ k}\Omega$) in parallel with S_1 , S_2 , and S_3 switches (Fig. 4a). Under a temperature difference of 8 K, the open-circuit voltage reached 360 mV with 'on' state only for S_1 switch. Then, a capacitor ($470 \mu\text{F}$) was charged to 350 mV by closing S_2 , and subsequently the capacitor was discharged by a load resistor (S_2 and S_3 were closed). After repeated charge/discharge cycles, voltage approached zero when the temperature of the device became uniform.

The output voltage was further boosted by connecting 2, 4, 6, and 8 units in series (Fig. 4b). Under the temperature difference of $7.6 \text{ }^\circ\text{C}$ and $3.9 \text{ }^\circ\text{C}$, serially joined eight modules produced 2.8 V and 1.45 V, respectively. The linear relationship between voltage generation and the number of modules indicates that the output voltage can be elevated by serial connection. Figure 4c displays that a digital hygrometer with a LCD screen was powered by four serially coupled modules with a $470 \mu\text{F}$ capacitor. When the hygrometer started to work, voltage and current were recorded. We also configured both current and voltage with combined parallel and serial circuits. We made a module consisting of serially joined four units, and two modules (m_1 and m_2) were hooked up in parallel along with a $470 \mu\text{F}$ capacitor, and the voltage profile of the two modules (closed S_1 , S_2 , and S_3) was compared with that of one module (m_1 only, S_1 and S_3 were closed) (Fig. 4d). Initially, the open-circuit voltage was identical for both circuits, but the capacitor was charged more rapidly

Deleted: is relatively rapid but saturated,

Deleted: ing

Deleted:

Deleted:

with two modules, showing ~2.5 min for one time constant (63%) compared to ~9 min with one module. This elucidates that our device offers options for optimizing voltage and current for a system of interest.

Based on the performance assessment, we have developed a battery-less and self-sustainable fever detector, which could be utilized for early detection and continuous monitoring of viral infectious diseases for humans and livestock. However, one-time measurements may suffer from false negative results because they could be strongly affected by human errors, environmental conditions, and stage of infection, in addition to a risk of cross infection. These drawbacks could be mitigated by continuous monitoring with low-cost and self-sustainable sensors and electrochromic display devices (Fig. S12 and Fig. 5c,d,e). To visualize the temperature changes, we integrated an electrochromic display with serially-connected four devices with the fully-developed oxidation layer. The electrochromic display was made of Prussian white (Fig. S13a). When electricity was supplied to the display, the color was changed from white to blue, and then, without electricity, from blue to white, reversibly. To mimic a situation with a fever, the heat flux from a human without and with a fever was assumed to be 360 W m^{-2} and 580 W m^{-2} , respectively, which have yielded the temperature differences (ΔT) of $1.8 \text{ }^\circ\text{C}$ and $2.9 \text{ }^\circ\text{C}$, respectively, across the device (Fig. 5a) (See Section 8 in SI⁴⁰⁻⁴³). Under the temperature differences, thermoelectric voltages of ~0.6 V and ~1 V were observed (Fig. 5b). Before the operation of the device, the current status window on the left in Fig. 5c is white. When ΔT was $1.8 \text{ }^\circ\text{C}$, the color of the window was slightly changed to light blue, indicating the temperature is below a fever on the reference bar (Fig. 5d). Further increase in ΔT to $2.9 \text{ }^\circ\text{C}$ resulted in a darker blue window, indicating fever or higher

- Deleted: characteristic
- Deleted: studies
- Deleted: the
- Deleted:
- Deleted: pandemics
- Deleted: (e.g., COVID, swine flu)

Deleted: As the most common symptom of viral infection leading to pandemic such as COVID, SARS, MERS, and swine flu is fever, temperature monitoring is possibly a promising and practical measure to identify potential infectees. For example, the majority of COVID patients are known to have had fever.

- Deleted: /
- Formatted: Superscript
- Deleted: /
- Formatted: Superscript
- Deleted: 9

temperature. For other or more sophisticated applications, the number of devices can be readily altered, and the color of the reference bar can be adjusted for desired temperature ranges.

In summary, we discovered a new method of generating an extremely large thermopower, -87 mV K^{-1} at 22 °C and 50% RH by utilizing the change in the corrosion potential due to the thermo-diffusion of water, through a series of systematic and rigorous experimental studies for unveiling the working mechanisms. We also substantiated our thermal energy harvester could be a viable option for powering electronic devices including a battery-less fever detector that can be distributed to many unspecified people at public places at an extremely low price. We anticipate that this study opens up and facilitates subsequent research for achieving even higher thermopower and developing self-sustainable electronic devices such as disposable, low-cost, and compact sensors due to battery-less operation.

Method

Materials and devices

Polystyrene sulfonic acid (PSS-H) (M.W. 75000, 30% wt%; Alfa Aesar), aniline (99+%; Alfa Aesar), hydrochloric acid (HCl, 36.5% - 38.0%, ACS; Macron Fine Chemicals), carbon steel shim (1008-1010 carbon steel, thickness: 0.005 inch; Precision Brand Products, Inc.), graphite foil (\geq 99.8% metals basis, thickness 0.254 mm; Alfa Aesar), iron(III) chloride (anhydrous; Sigma Aldrich), Iron(II) chloride (tetrahydrate, 98%; Alfa Aesar), iron foil (iron \geq 99.99% metals basis, thickness 0.1 mm; Alfa Aesar), ammonium persulfate (ACS; J.T. Baker), deionized (DI) water ($>$ 18 M Ω cm^{-1}), potassium ferricyanide ($K_3Fe(CN)_6$, 98.5%; Acros Organics), tetraethylammonium

Deleted: -

Deleted: /K

Deleted: without batteries

Formatted: Font: Bold

Formatted: Superscript

perchlorate (TEAP, 98%; Alfar Asear), ITO glass ($100 \Omega \text{ sq}^{-1}$; Nanocs), Nafion 115 membrane (Fuel Cell Earth), hydrazine (Anhydrous, 98%; Sigma Aldrich). The digital hygrometer with LCD screen was purchased from Linkstyle.

Material synthesis

The following procedure was used to synthesize the polyelectrolyte made of polyaniline and polystyrene sulfonate (PANI:PSS)³⁰. First, 30-wt% PSS-H (30 g) and aniline (4 mL) were added to DI water (80 mL), and the solution was stirred for 1 h. Ammonium persulfate solution was diluted in DI water (50 mg of ammonium persulfate per mL of water). Then, 40 mL of the aqueous ammonium persulfate solution was slowly dropped into the PANI:PSS solution for 30 min using a syringe pump while the solution was stirred using a magnetic bar. The solution was stirred for 5 min and then stored overnight. Subsequently, the solution was poured into acetone (1 L) for precipitation, and then decanted the solution. The collected precipitate were washed by acetone five times, and then dried at $50 \text{ }^\circ\text{C}$ in an oven to obtain dark green PANI:PSS (~13 g).

The dried PANI:PSS precipitate dissolved in DI water (20 wt%) by stirring for 2 h, and then the hydrochloric acid was added to the PANI:PSS solution (10 vol% of the hydrochloric acid in the PANI:PSS solution). After the solution was stirred for 1 h, the solution (~3 g) was drop-casted on two carbon steel foils (dimension is 2 cm by 2 cm), and two pieces were put together before they were fully dried (typically after 12 h in a fume hood). The sample was left in a fume hood for 5 days at room temperature. During the drying process, carbon paint was coated on the outer side of the carbon steel for good electrical connection with lead metal tab for electrical measurements.

Formatted: Superscript

The thickness of the device was ~2 mm.

Thermopower measurement

Thermopower was measured using our custom-built setup (Fig. S3) in the humidity-controlled chamber. Peltier devices with aluminum block were used for controlling temperature. The device was placed in the middle of two Peltier devices to control the temperature difference. Two thermocouples were placed on top and bottom of the device to measure the temperature while two copper wires were used to measure the voltage between the two electrodes. Upon altering the current passing through the Peltier device, the temperature difference was varied. We took data after sample was left in a particular relative humidity level for at least 8 h to ensure the sample reached steady state. Voltage as a function of time was measured at various temperature differences, typically 6-8 points, and the linear slope in temperature difference (x-axis) vs. voltage (y-axis) was sought for thermopower. The temperature difference was also created by one Peltier device as a heater and an aluminum block as a heat sink.

Electrochemical impedance spectroscopy (EIS) measurement

For EIS measurements, our device was kept under different humidity levels for 8 hours, and then it was scanned over a frequency range between 0.1~10⁶ Hz.

Data availability

The data that support the findings of this study are available from the corresponding author upon

Deleted: ¶

Formatted: Font: Bold

reasonable request.

Moved down [1]: Acknowledgement ¶

The authors acknowledge financial supports from U.S. National Science Foundation (CBET 1805963).¶

Formatted: Font: Bold

References

- 1 Mao, J. *et al.* Advances in thermoelectrics. *Adv. Phys.* **67**, 69-147 (2018).
- 2 Wang, H. & Yu, C. Organic thermoelectrics: materials preparation, performance optimization, and device integration. *Joule* **3**, 53-80 (2019).
- 3 Zhao, L.-D. *et al.* Ultrahigh power factor and thermoelectric performance in hole-doped single-crystal SnSe. *Science* **351**, 141-144 (2016).
- 4 He, W. *et al.* High thermoelectric performance in low-cost SnS_{0.91}Se_{0.09} crystals. *Science* **365**, 1418-1424 (2019).
- 5 Kang, S. D. & Snyder, G. J. Charge-transport model for conducting polymers. *Nat. Mater.* **16**, 252-257 (2017).
- 6 Kim, S. I. *et al.* Dense dislocation arrays embedded in grain boundaries for high-performance bulk thermoelectrics. *Science* **348**, 109-114 (2015).
- 7 Russ, B., Glaudell, A., Urban, J. J., Chabinyk, M. L. & Segalman, R. A. Organic thermoelectric materials for energy harvesting and temperature control. *Nat. Rev. Mater.* **1**, 16050 (2016).
- 8 Kim, S. L., Lin, H. T. & Yu, C. Thermally chargeable solid-state supercapacitor. *Adv. Energy Mater.* **6**, 1600546 (2016).
- 9 Akbar, Z. A., Jeon, J.-W. & Jang, S.-Y. Intrinsically self-healable, stretchable thermoelectric materials with a large ionic Seebeck effect. *Energy Environ. Sci.* **13**, 2915-2923 (2020).
- 10 Kim, S. L., Hsu, J.-H. & Yu, C. Intercalated graphene oxide for flexible and practically large thermoelectric voltage generation and simultaneous energy storage. *Nano Energy* **48**, 582-589 (2018).
- 11 Kim, S. L., Hsu, J.-H. & Yu, C. Thermoelectric effects in solid-state polyelectrolytes. *Org. Electron.* **54**, 231-236 (2018).
- 12 Jiao, F. *et al.* Ionic thermoelectric paper. *J. Mater. Chem. A* **5**, 16883-16888 (2017).
- 13 Kim, B., Hwang, J. U. & Kim, E. Chloride transport in conductive polymer films for an n-type thermoelectric platform. *Energy Environ. Sci.* **13**, 859-867 (2020).
- 14 Han, C.-G. *et al.* Giant thermopower of ionic gelatin near room temperature. *Science* **368**, 1091-1098 (2020).
- 15 Li, T. *et al.* Cellulose ionic conductors with high differential thermal voltage for low-grade heat harvesting. *Nat. Mater.* **18**, 608-613 (2019).
- 16 Xue, J. *et al.* Vapor-activated power generation on conductive polymer. *Adv. Funct. Mater.* **26**, 8784-8792 (2016).
- 17 Zhao, F., Liang, Y., Cheng, H., Jiang, L. & Qu, L. Highly efficient moisture-enabled electricity generation from graphene oxide frameworks. *Energy Environ. Sci.* **9**, 912-916 (2016).
- 18 Zhao, F., Wang, L., Zhao, Y., Qu, L. & Dai, L. Graphene Oxide Nanoribbon Assembly toward Moisture-Powered Information Storage. *Adv. Mater.* **29**, 1604972 (2017).
- 19 Lee, S., Eun, J. & Jeon, S. Facile fabrication of a highly efficient moisture-driven power generator using laser-

- induced graphitization under ambient conditions. *Nano Energy* **68**, 104364 (2020).
- 20 Xu, T. *et al.* An efficient polymer moist-electric generator. *Energy Environ. Sci.* **12**, 972-978 (2019).
- 21 Liang, Y. *et al.* Electric power generation via asymmetric moisturizing of graphene oxide for flexible, printable and portable electronics. *Energy Environ. Sci.* **11**, 1730-1735 (2018).
- 22 Zhao, F., Cheng, H., Zhang, Z., Jiang, L. & Qu, L. Direct power generation from a graphene oxide film under moisture. *Adv. Mater.* **27**, 4351-4357 (2015).
- 23 Kim, B. *et al.* Robust high thermoelectric harvesting under a self-humidifying bilayer of metal organic framework and hydrogel layer. *Adv. Funct. Mater.* **29**, 1807549 (2019).
- 24 Choi, K., Kim, S. L., Yi, S.-i., Hsu, J.-H. & Yu, C. Promoting dual electronic and ionic transport in PEDOT by embedding carbon nanotubes for large thermoelectric responses. *ACS Appl. Mater. Interfaces* **10**, 23891-23899 (2018).
- 25 Kang, T. J. *et al.* Electrical power from nanotube and graphene electrochemical thermal energy harvesters. *Adv. Funct. Mater.* **22**, 477-489 (2012).
- 26 Lee, S. W. *et al.* An electrochemical system for efficiently harvesting low-grade heat energy. *Nat. Commun.* **5**, 3942 (2014).
- 27 Yu, B. *et al.* Thermosensitive crystallization-boosted liquid thermocells for low-grade heat harvesting. *Science*, abd6749 (2020).
- 28 Jang, J., Ha, J. & Cho, J. Fabrication of water-dispersible polyaniline-poly(4-styrenesulfonate) nanoparticles for inkjet-printed chemical-sensor applications. *Adv. Mater.* **19**, 1772-1775 (2007).
- 29 Mackay, A. L. β -Ferric oxyhydroxide. *Mineral Mag* **32**, 545-557 (1960).
- 30 Li, Y., Ying, B., Hong, L. & Yang, M. Water-soluble polyaniline and its composite with poly (vinyl alcohol) for humidity sensing. *Synth Met* **160**, 455-461 (2010).
- 31 Cheng, H., He, X., Fan, Z. & Ouyang, J. Flexible quasi-solid state ionogels with remarkable Seebeck coefficient and high thermoelectric properties. *Adv. Energy Mater.* **9**, 1901085 (2019).
- 32 Zhao, D. *et al.* Polymer gels with tunable ionic Seebeck coefficient for ultra-sensitive printed thermopiles. *Nat. Commun.* **10**, 1093 (2019).
- 33 Yang, P. *et al.* Wearable thermocells based on gel electrolytes for the utilization of body heat. *Angew. Chem. Int. Ed.* **128**, 12229-12232 (2016).
- 34 Bonetti, M., Nakamae, S., Roger, M. & Guenoun, P. Huge Seebeck coefficients in nonaqueous electrolytes. *J. Chem. Phys.* **134**, 114513 (2011).
- 35 Jiang, Q. *et al.* High thermoelectric performance in n-type perylene bisimide induced by the Soret effect. *Adv. Mater.*, 2002752 (2020).
- 36 Cheng, F. *et al.* FTIR analysis of water structure and its influence on the flotation of arcanite (K_2SO_4) and epsomite ($MgSO_4 \cdot 7H_2O$). *Int. J. Miner. Process.* **122**, 36-42 (2013).
- 37 Stan, C. S., Popa, M., Olariu, M. & Secula, M. S. Synthesis and characterization of PSSA-polyaniline composite with an enhanced processability in thin films. *Open Chem.* **13**, 467-476 (2015).
- 38 Gardiner, C. & Melchers, R. Corrosion of mild steel in porous media. *Corros Sci* **44**, 2459-2478 (2002).
- 39 Yan, M., Sun, C., Xu, J. & Ke, W. Anoxic corrosion behavior of pipeline steel in acidic soils. *Ind. Eng. Chem. Res.* **53**, 17615-17624 (2014).
- 40 Stettler, N., Schutz, Y., Whitehead, R. & Jéquier, E. Effect of malaria and fever on energy metabolism in Gambian children. *Pediatr. Res.* **31**, 102-106 (1992).
- 41 Du Bois, E. F. The mechanism of heat loss and temperature regulation. *Ann. Intern. Med.* **12**, 338-395 (1938).

42 de Rivera, P. J. R., de Rivera, M. R., Socorro, F. & de Rivera, M. R. Measurement of human body surface heat flux using a calorimetric sensor. *J. Therm. Biol.* **81**, 178-184 (2019).

43 Herzog, L. W. & Coyne, L. J. What is fever? Normal temperature in infants less than 3 months old. *Clin Pediatr* **32**, 142-146 (1993).

Acknowledgement

The authors acknowledge financial supports from U.S. National Science Foundation (CBET 1805963).

Author contributions

C. Y. and Y. Z. conceived the idea, carried out the experiments and analyses, and wrote the manuscript. A. S. and A. C. assisted the experiments.

Ethics declarations

Competing interests

The authors declare no competing interests.

Moved (insertion) [1]

Formatted: Font: Bold

Formatted: Font: Bold

Formatted: Font: Bold

Figure 1. (a) Device fabrication procedure. Scanning electron micrographs of (b) the carbon steel electrode after detaching PANI:PSS, (c) a porous area, and (d) a flat area. (e) X-ray diffraction data for a pristine carbon steel, the electrode shown in 'b', and reference data for β-FeOOH²⁹. (f) Water uptake of PANI:PSS after the sample was left under the relative humidity for 8 h. Error bar: one standard deviation.

Figure 2. (a) Voltage profile of the device with the fully-developed oxidation layer under 50% RH as a function of time when the temperature difference (ΔT) was altered. The inset shows the saturated voltage at the corresponding temperature to seek the slope (93 mV K^{-1}). Note that thermopower has the opposite sign (-93 mV K^{-1}). (b) Thermopower (absolute value) of our devices with the intermediate (IM) and fully-developed (FD) oxidation layers along with data in the literature, Kim^a, Han¹⁴, Kim^b, Jiao¹², Akbar⁹, Kim^c, Kim^d, Cheng³¹, Kim^e, Kim^f, Kim^g, Zhao^a, Zhao^b, and Yang³³. Error bar: one standard deviation. ¹RH is assumed to be 50%.

Deleted: as
 Deleted: /K
 Deleted: /K
 Deleted: for thermopower.

Figure 3. (a) Experimental procedure to identify the influence of water uptake on the corrosion potential of carbon steel. (b) Voltage and change of water mass when the top electrode was left under different RH following the procedure in 'a'. (c) ATR-FTIR spectra of PANI:PSS from the colder side as the temperature difference (ΔT) was varied. The inset illustrates the in-situ ATR-FTIR experiment configuration. (d) Evans diagram shows the cathodic reduction reaction (black line), the anodic oxidation reaction (green line) under uniform temperature. The green line shifts towards the red (or blue) line with less (or more) water on the electrodes. (e-h) Illustration showing thermoelectric voltage generation and corresponding potential changes in the hotter and colder sides. (e) Uniformly distributed water in PANI:PSS under $\Delta T=0$. (f) As $\Delta T>0$, water molecules migrate from the hotter to colder side. (g) Voltage generation at steady-state under $\Delta T>0$. (h) Water molecules return to their initial distributed states under $\Delta T \rightarrow 0$.

Figure 4. (a) Voltage generation from one device under $\Delta T=8^\circ\text{C}$ followed by charging/discharging a capacitor. (b) Linearly increasing voltage with more number of devices connected in series under $\Delta T = 3.9$ and 7.6°C . The upper inset is the voltage profile as a function of time as 2, 4, 6, and 8 devices were joined in series. The lower inset depicts two devices connected in series. (c) Voltage and current profiles when the capacitor ($470\ \mu\text{F}$) was hooked up to the digital hydrometer shown in the inset. (d) Voltage profiles for one module (m_1) and two modules connected in parallel (m_1 and m_2), followed by charging a capacitor ($470\ \mu\text{F}$) as a function of time under $\Delta T = 7.6^\circ\text{C}$. Each module consists of 4 devices connected in series.

Figure 5. (a) Temperature difference as a function of time when heat is dissipated at the rates of $360 \text{ W}\cdot\text{m}^{-2}$ and $580 \text{ W}\cdot\text{m}^{-2}$, to mimic normal and fever conditions, respectively. The inset illustrates the temperature measurement configuration with a heater. (b) Voltage profile as a function of time from serially connected four devices under the temperature differences of 1.8 and 2.9 °C. The inset illustrates the thermal energy harvester was directly connected to the fever detector. (c) A photograph of the fever detector with an electrochromic display window showing the current status and a reference color bar when heat is not supplied to the device ($\Delta T=0^\circ\text{C}$). (d) When $\Delta T\approx 1.8^\circ\text{C}$, the color in the status window became light blue, indicating the device was working under a normal condition. (e) When $\Delta T\approx 2.9^\circ\text{C}$, the status window displayed dark blue above the fever color in the reference bar.

Deleted: /
Formatted: Superscript
Deleted: /
Formatted: Superscript

Reviewer comments, second round –

Reviewer #1 (Remarks to the Author):

The manuscript reports a large thermopower with the change of humidity and temperature. And it is also supported by adequate experimental data. But there are some key issues that we cannot support this manuscript to publish on Nat. Comm..

1. The author provides a lot of experimental data to prove the mechanism of their device. In our opinion, their explanation is reasonable. However, we can't agree with the term of "thermopower" to describe their system. This is not a thermoelectric effect, and the output of electric energy is not transformed from heat energy.
2. The author's viewpoint about "battery-less" is wrong. The corrosion of the carbon steel is certainly a battery discharge process. This can be called a simple long-life battery, but not "battery-less".
3. In our opinion, the authors used two water-sensitive batteries, connected by PANI:PSS. The water content difference between the two batteries was achieved by PANI: PSS under temperature gradient, leading to a potential difference eventually. This is out of the concept of thermoelectricity, and it is not appropriate to compare with other thermoelectric systems.

Reviewer #3 (Remarks to the Author):

The authors have appropriately addressed all my concerns and the manuscript has been sufficiently improved. It is my opinion that the paper is suitable for publication due to the detailed investigation of the thermopower origin combined with this being the first report of such a large thermopower arising from thermally driven water concentration gradients.

Rebuttal for Manuscript ID: NCOMMS-21-07317

REFEREE 1

Reviewer's comments

The manuscript reports a large thermopower with the change of humidity and temperature. And it is also supported by adequate experimental data. But there are some key issues that we cannot support this manuscript to publish on Nat. Comm..

1. The author provides a lot of experimental data to prove the mechanism of their device. In our opinion, their explanation is reasonable. However, we can't agree with the term of "thermopower" to describe their system. This is not a thermoelectric effect, and the output of electric energy is not transformed from heat energy.

Authors' responses

We truly appreciate your valuable time and efforts in reviewing this manuscript.

We agree that the working principle in our work is different from those of conventional thermoelectric systems. We simply used "thermoelectric" because we couldn't find other suitable alternative words at the time of submission. After careful consideration, we came up with a new term, "thermo-hydro-electrochemical", which reflects what happens in our system. We have removed "thermoelectric" from the title, and used "thermo-hydro-electrochemical" instead, which clearly states that the mechanism originates from hybridized effects. The new title is "Colossal Thermo-Hydro-Electrochemical Voltage Generation for Self-sustainable Operation of Electronics".

Reviewer's comments

2. The author's viewpoint about "battery-less" is wrong. The corrosion of the carbon steel is certainly a battery discharge process. This can be called a simple long-life battery, but not "battery-less".

Authors' responses

Thank you for the comment. We have removed "battery-less" from the title to avoid misconception.

Reviewer's comments

3. In our opinion, the authors used two water-sensitive batteries, connected by PANI:PSS. The water content difference between the two batteries was achieved by PANI: PSS under temperature gradient, leading to a potential difference eventually. This is out of the concept of thermoelectricity, and it is not appropriate to compare with other thermoelectric systems.

Authors' responses

We are comparing voltage generation per temperature difference (i.e., thermally-induced voltage per Kelvin), regardless of the working principle. The traditional thermoelectric effect is also different from newly reported ionic thermoelectric effects based on the Soret effect (thermo-diffusion of ions) and thermogalvanic cells. Here we have changed the word, thermopower in order to clearly show we are comparing thermally-induced voltage per Kelvin rather than the conventional Seebeck coefficient. We revised Figure 2b, stating "Thermal-to-Electrical Conversion Factor" (or simply TtoE factor) instead of thermopower.

REFeree 3

Reviewer's comments

The authors have appropriately addressed all my concerns and the manuscript has been sufficiently improved. It is my opinion that the paper is suitable for publication due to the detailed investigation of the thermopower origin combined with this being the first report of such a large thermopower arising from thermally driven water concentration gradients.

Authors' responses

We truly appreciate your valuable time and efforts in reviewing this manuscript. We also appreciate your opinion to improve this manuscript.

Colossal **Thermo-Hydro-Electrochemical Voltage Generation** for

Self-sustainable Operation of Electronics

Yufan Zhang¹, Ahrum Sohn², Anirban Chakraborty², Choongho Yu^{1,2} *

Deleted: Thermal-to-Electrical Energy ConversionThermopower

Deleted: Battery-less and

Deleted: ¶

¹ Department of Materials Science and Engineering, Texas A&M University, College Station, Texas, 77843 USA

² Department of Mechanical Engineering, Texas A&M University, College Station, Texas, 77843 USA

*Corresponding author: chyu@tamu.edu

Keywords

Thermal-to-Electrical Energy Conversion; Thermoelectric; Corrosion; Energy harvesting;

Thermo-diffusion

Deleted: Thermopower

Abstract

Thermoelectrics are suited to converting dissipated heat into electricity for operating electronics, but the small voltage ($\sim 0.1 \text{ mVK}^{-1}$) from the Seebeck effect has been problematic. Here a new approach with thermo-hydro-electrochemical effects can generate the largest ever reported thermal-to-electrical energy conversion factor (TtoE factor), -87 mVK^{-1} , with low-cost carbon steel electrodes and a solid-state polyelectrolyte made of polyaniline and polystyrene sulfonate (PANI:PSS). We discovered that the thermo-diffusion of water in PANI:PSS under a temperature gradient induced less (or more) water on the hotter (or colder) side, raising (or lowering) the corrosion overpotential in the hotter (or colder) side and thereby generating output power between the electrodes. The practicality of the colossal TtoE factor has been validated by developing a fever indicator with typically dissipated thermal energy from a human body. Our new findings are expected to facilitate subsequent research for further increasing TtoE factor and utilizing dissipated thermal energy.

Deleted:

Deleted: al-and-

Deleted: energy hybrid

Deleted: (TtoE factor)

Deleted:

Deleted: in a typical ambient condition

Deleted: two components,

Deleted: thermopower

Deleted: only

Deleted: , showing color changes with and without fever reversibly

Deleted: thermopower

Introduction

Thermoelectricity refers to converting heat to electricity or vice versa, and has been used for various applications including thermocouples and Peltier devices. It is desired to generate a large voltage per temperature difference (*i.e.*, large thermopower or Seebeck coefficient in the unit of V K^{-1}), resulting from the thermo-diffusion of electrons called the Seebeck effect¹. As for traditional solid-state inorganic and recent organic thermoelectric materials, small thermopower values on the order of $0.01\sim 0.1 \text{ mV K}^{-1}$ near room temperature have been one of the major hurdles in acquiring high device performances²⁻⁷. In the last several years, the thermo-diffusion of ions, called the Soret effect,⁸ has been utilized to aggrandize thermopower to 8 mV K^{-1} with polystyrene sulfonic acid due to the substantial difference in thermo-diffusion between mobile protons and immobile anions⁹. While it has been difficult to have greater than several mV K^{-1} , several recent studies yielded gigantic thermopower¹⁰⁻¹³ including 18 mV K^{-1} from the migration of chloride ions in n-type mixed ionic–electronic conductive polymer composite films¹⁴, 17 mV K^{-1} partially from KCl in a quasi-solid-state ionic thermoelectric material¹⁵, and 24 mV K^{-1} with sodium ions in cellulose ionic conductors¹⁶. It is worth noting that the thermally-induced voltage is different from moisture-powered generators (MPG) based on ions movement carried by water-diffusion and ion accumulation¹⁷⁻²³. MPG works only when environmental humidity changes or water droplets are applied to the device, and it does not respond to a temperature difference.

To induce a thermally-induced voltage on the order of $1\sim 10 \text{ mV K}^{-1}$, a few different mechanisms have been recently reported, including the thermo-diffusion of electron/ion mixture (both Soret and Seebeck effects)^{24,25} and temperature-dependent redox reactions with redox

couples in liquid states²⁶⁻²⁸. It appears that solid-state polyelectrolytes utilizing the Soret effect are the best option for high thermopower so far, and their highest thermopower values were obtained at unusually high (70-100%) relative humidity (RH) rather than typical room humidity (~50% RH). Water is an electrolyte for the ions, improving their mobility, and water makes mobile ions readily dissociated from their counter ions^{9,12,24}. However, high water uptake in solid-state polyelectrolytes often causes stability problems due to irreversible water evaporation and swelling. Considering high thermal-to-electrical conversion (TtoE) factor is the key to the performance, it is valuable to seek other routes for attaining even bigger TtoE factors. It should be noted that we used a new term, TtoE factor rather than thermopower and Seebeck coefficient in order to simultaneously account for various principles generating thermally-induced voltage. For example, to acquire a working voltage (> 1 V) of typical wearable electronics with traditional inorganic materials, at least 1000 thermoelectric legs should be serially connected under a temperature difference of 10 °C.

Here we report a new approach based on the variation of potentials caused by a temperature difference. We used readily available carbon steel as electrodes to obtain a colossal TtoE factor of -87 mV K⁻¹ under a typical ambient condition (50% RH, 22 °C). Porous hydrophilic layers were formed on the carbon steel, and a hygroscopic solid-state polyelectrolyte layer was used between the two electrodes. We discovered that, upon imposing a temperature difference, the thermo-diffusion of water from the hotter side to the colder side altered the water uptake in the electrode, differentiating the potential of the two electrodes. Based on the new mechanism, a self-sustainable fever detection device has been operated as a proof of principle, which could be helpful in the early

Deleted: thermopower

Deleted: of thermoelectric applications

Deleted: thermopower

Deleted: the

Deleted: thermopower

Deleted: battery-less,

and fast detection of fever commonly observed from a viral infection such as COVID, SARS, MERS, and swine flu pandemic. The following include fabrication and characterization of materials and devices, investigation of working principles, and applications.

Results and discussion

Our device consists of polyaniline and polystyrene sulfonate (PANI:PSS) as a solid-state electrolyte and carbon steel foils as electrodes (Fig. 1a). PANI:PSS powders (Fig. S1 for FTIR) were synthesized with polystyrene sulfonic acid (PSS-H) and aniline ²⁹, and then they were dissolved in deionized water with hydrochloric acid. The solution was drop-casted on two carbon steel electrodes, and two pieces were assembled before they were fully dried. After the assembly, the sample was left in a fume hood. During this time period, the surface of the carbon steel was corroded, forming a new layer between PANI:PSS and electrodes, as shown in Fig. 1b. We observed a porous layer composed of few micron long nanorods (Fig. 1c) under a flat PANI:PSS layer (Fig. 1d). The XRD patterns (Fig. 1e) indicate that the nanorod is made of β -FeOOH ³⁰. As PANI:PSS is a hygroscopic material ^{9,31}, the water uptake is a strong function of RH in the environment. The amount of water soaked in PANI:PSS under different RH was characterized as a function of time (Fig. S2), and steady-state values are summarized in Fig. 1f. The water uptake in the sample was augmented with higher RH, and was found to be ~15 wt% in PANI:PSS under a typical room environment (50% RH). Transport property measurements were carried out after the water uptake reached steady states.

For thermally-induced voltage measurements, the temperature difference between two electrodes was varied up to ± 6 K, and voltage was recorded as a function of time. Figure 2a shows

the generated voltage of the devices with 15% water uptake (RH=50%), and the saturated voltage as a function of temperature difference was plotted in the inset of Fig. 2a. Those of all the other samples are shown in Fig. S3, Table S1, and Table S2. The slopes from the linear fitting are the absolute values of the \downarrow TtoE factor, which are plotted against RH along with those from various thermal-to-electrical energy conversion principles in the literature (Fig. 2b)^{9,10,12-15,24,32-34}. It should be noted that the sign of the number for the slope should be reversed to get the \downarrow TtoE factor (i.e., a positive slope means a negative \downarrow TtoE factor) like conventional thermopower. We found that magnitude of the \downarrow TtoE factor gets bigger as we elongated the oxidation time of the carbon steel in the ambient condition, but it did not further increase after ~60 days. We observed consistent values, -85 ~ -87 mV K⁻¹ after 120 days and 180 days. The two different cases from fully-developed (60 days) and intermediate (14 days) oxidation layers are shown in Fig. 2b. At 30% RH, the \downarrow TtoE factor of the fully-developed case was -48 mV K⁻¹, and, under 50% RH, it was boosted up to -87 mV K⁻¹, which is the highest value ever reported, to the best of our knowledge. Even for the intermediate case, -47 mV K⁻¹ at 50% RH in our thermo-hydro-electrochemical hybrid device is much higher than other thermally-induced voltage in the literature.

The difference in the \downarrow TtoE factor for the two cases mainly comes from the impedance change of the oxidation layer. A greater potential difference between electrodes can be developed when the impedance of the oxidation layer was enlarged. According to the electrochemical impedance spectroscopy results (Fig. S4), the impedance of PANI:PSS on the order of 10 Ω was significantly raised to values on the order of k Ω with the oxidation layers, and larger impedance was observed from the fully-developed cases. It is interesting to see the distinct humidity dependency from our

Deleted: thermopower

Deleted:

Deleted: thermopower

Deleted: thermopower

Deleted: thermopower

Deleted: thermopower

Deleted: thermopower

Deleted: thermopower

Deleted: al-and-chemical energy

Deleted: those

Deleted: the

Deleted: thermopower

sample where there is an optimum RH while the others show monotonically increasing T_{toE} factors with RH. In fact, the optimum performance at 50% RH is ideal because it is close to that of typical indoor environments. On the other hand, this would be an indicator that the working principle of our system is different from the others.

In the literature reporting high T_{toE} factors, the thermo-diffusion of ions (e.g., proton) was identified to be the main driver^{8,9,11-16,35,36}. Here, to test the influence of the thermo-diffusion of ions in PANI:PSS on the T_{toE} factor, we assembled a cell with graphite foils instead of carbon steel (Fig. S5a). We found that the slope in temperature difference vs. voltage plot in Fig. S5b is opposite to that of our device in the inset of Fig. 2a. When protons in PANI:PSS migrate from the hotter side to the colder side, a negative slope (or a positive T_{toE} factor) was obtained. Moreover, the T_{toE} factor from the device with graphite electrodes was found to be only 0.97 mV K^{-1} , which is much smaller than -87 mV K^{-1} from our carbon steel based device. Therefore it is clear that the working principle of our device is different from those in the literature. Instead, the graphite device is similar to MPG¹⁷⁻²³ because they both rely on proton migration. To verify the difference between MPG and ours, we carried out experiments exposing moisture instead of heat to one of the electrodes (Fig. S6). PANI:PSS with our carbon steel electrodes generated $\sim 360 \text{ mV}$ under RH difference of 58%, whereas the graphite electrode device generated only $\sim 0.2 \text{ mV}$. We also tested the influence of $\beta\text{-FeOOH/Fe}^{2+}$ redox couple on the T_{toE} factor by sandwiching $\beta\text{-FeOOH}$ between PANI/PSS and graphite foils. We observed voltage continuously changed despite constant temperature difference with a largest absolute value of $\sim 1.3 \text{ mV}$ under 4 K difference (Fig. S7). The large T_{toE} factor can be obtained only with the steel electrodes, suggesting a corrosion process

Deleted: thermopower

Deleted: thermopower

Deleted: thermopower

Deleted: thermopower

Deleted: thermopower

Deleted: thermopower

Deleted: thermopower

plays a key role in our system.

Then we investigated if corrosion was caused by the thermo-diffusion of protons in a 3-electrode configuration with the FeOOH-coated carbon steel electrode (taken out from the device) as a working electrode (Fig. S8). Voltage sweeping between a Pt counter and reference electrodes resulted in Tafel curves, showing a corrosion potential of -0.45 V vs. Ag/AgCl when the current between the working and counter electrodes approached zero. As HCl was gradually added to have pH of 1, 0.8, and 0.6 (initially pH = 1.25), the potential shifted towards positive values with more protons (or a lower pH). This result denotes more protons make the corrosion potential of carbon steel more positive, which is the same as Fig. S5 where negative voltages under $\Delta T > 0$ (protons on the bottom electrode) were observed. This behavior is opposite to the trend of our device in Fig. 2a. A similar experiment (Fig. S9) also exhibited that voltage was shifted more positively with the addition of protons, confirming that the thermo-diffusion of the proton is not the major contributor in our system. Furthermore, our experimental results in Fig. S10 proved that the corrosion potential in the Tafel curves is not a significant function of temperature.

As voltage generation strongly depends on humidity, we carried out experiments directly showing the influence of water uptake on voltage generation. One of the electrodes in the device was taken out of the device and exposed to environmental conditions whose RH was altered from 50% RH to 20% RH and 70% RH for 12 hours (Fig. 3a). The porous layer on the electrode can accommodate water from the humid environment or release water initially present in the 50% RH condition, as indicated by the mass change in drier and wetter conditions (Fig. 3b). When the electrode was re-assembled, the voltage was remarkably altered, showing higher (or lower)

potential with less (or more) water. When water moves from the hotter side to the colder side, the hotter side has a higher potential than the colder side, which agrees with the trend (slope) shown in our system (inset of Fig. 2a).

We used in-situ attenuated total reflectance (ATR) Fourier transform infrared spectroscopy (FTIR) to identify the thermo-diffusion of water in PANI:PSS by comparing the intensity of O-H stretching band for water,³⁷ which appears over a broad range near 2800~3700 cm⁻¹ with its peak³⁸ at ~3450 cm⁻¹ while one side of PANI:PSS was being heated (Fig. 3c). All the spectra were normalized by the peak intensity at 2914 cm⁻¹ corresponding to C-H stretching of the CH₂ group in PSS³⁹. The absorbance peak near 3450 cm⁻¹ from the colder side was intensified as the temperature difference was enlarged, suggesting water diffused from the hotter to the colder side. It should be noted that the colder side of the sample was in contact with the FTIR apparatus at room temperature to avoid water condensation, and a temperature lower than room temperature was not applied due to the risk of changes in the water content.

The Evans diagram in Fig. 3d explains how the voltage was developed. Initially, the corrosion potential is located at the intersection between the oxidation and reduction Tafel curves (green and black lines in Fig. 3d, respectively). Under a temperature gradient, the thermo-diffusion of water reduces the amount of water in the hotter side while that in the colder side increases. Water is an electrolyte in corrosion reactions, so a reduction in water renders the corrosion overpotential higher. Then the Tafel curve for oxidation shifts counterclockwise (red line in Fig. 3d), and then a new potential (crossover point) is established due to the following relation⁴⁰.

$$E_i^0 = E_{\text{anode}} + I_i R_i \quad (1)$$

Deleted: rmoelectric

E_i^0 and E_{anode} are the potential of corrosion and anodic reaction, respectively. $I_i R_i$ is the overpotential. I_i is the corrosion current and R_i is the resistance of the electrolyte, where the index i is either the hotter (h) or colder side (c). On the other hand, higher water uptake decreases the overpotential, lowering the crossover point (blue line in Fig. 3d). The newly established two crossover points between the reduction line and the raised/lowered oxidation lines for the hotter/colder sides create a potential difference between the two electrodes, as follows.

$$\Delta E = E_h^0 - E_c^0 = I_h R_h - I_c R_c \quad (2)$$

The potential difference is functions of the corrosion current and resistance, which are strongly affected by the amount of water in the electrodes ⁴¹.

Figure 3e-f illustrate water diffusion and corresponding electrode potentials. Under a uniform temperature, water is homogeneously distributed in PANI:PSS, and the potentials of the top and bottom electrodes are identical (Fig. 3e). When a temperature gradient is created, the water molecules in the hotter side diffuse to the colder side (Fig. 3f). As the amount of migrated water proliferates with time, the potential in the hotter side is raised while that in the colder side is lowered, escalating the potential difference between the two electrodes (Fig. 3g). When the temperature becomes uniform, the water in the colder side is re-distributed, and thereby the potential difference converges to zero (Fig. 3h). This working mechanism also explains the peak TtoE factor at 50% RH rather than monotonically ~~increasing trends~~, with a higher RH in the literature. When the water uptake in PANI:PSS is too high, it is hard to induce a significant difference in the water concentrations on the two electrodes, resulting in a lower voltage. Conversely, low water uptake is unfavorable to the thermo-diffusion of water due to the limited

- Deleted: thermopower
- Deleted: augmented
- Deleted: thermopower

amount of water ¹².

Our device is promising for various applications including sensors and energy harvesters. For example, the colossal T -to- E factor from this study, which is several orders of magnitude larger than those of thermocouples, could give a substantial voltage response to a temperature difference and thereby ameliorate the signal-to-noise ratio. It is worth mentioning that the corrosion of the carbon steel resulted in only ~ 18 μm reduction for 6 months (Fig. S11a). Even if we assume continuous dissolution of carbon steel, 0.4-mm thick carbon steel could last longer than 10 years. Here we first demonstrated its functionality as an energy harvester using the device with the intermediate oxidation layer. A single unit device was connected to a capacitor (470 μF) and a load resistor (10 $\text{k}\Omega$) in parallel with S_1 , S_2 , and S_3 switches (Fig. 4a). Under a temperature difference of 8 K, the open-circuit voltage reached 360 mV with 'on' state only for S_1 switch. Then, a capacitor (470 μF) was charged to 350 mV by closing S_2 , and subsequently the capacitor was discharged by a load resistor (S_2 and S_3 were closed). After repeated charge/discharge cycles, voltage approached zero when the temperature of the device became uniform.

The output voltage was further boosted by connecting 2, 4, 6, and 8 units in series (Fig. 4b). Under the temperature difference of 7.6 $^\circ\text{C}$ and 3.9 $^\circ\text{C}$, serially joined eight modules produced 2.8 V and 1.45 V, respectively. The linear relationship between voltage generation and the number of modules indicates that the output voltage can be elevated by serial connection. Figure 4c displays that a digital hygrometer with a LCD screen was powered by four serially coupled modules with a 470 μF capacitor. When the hygrometer started to work, voltage and current were recorded. We also configured both current and voltage with combined parallel and serial circuits. We made a

Deleted: thermopower

Deleted: c

Deleted: thermopower

module consisting of serially joined four units, and two modules (m_1 and m_2) were hooked up in parallel along with a 470 μF capacitor, and the voltage profile of the two modules (closed S_1 , S_2 , and S_3) was compared with that of one module (m_1 only, S_1 and S_3 were closed) (Fig. 4d). Initially, the open-circuit voltage was identical for both circuits, but the capacitor was charged more rapidly with two modules, showing ~ 2.5 min for one time constant (63%) compared to ~ 9 min with one module. This elucidates that our device offers options for optimizing voltage and current for a system of interest.

Based on the performance assessment, we have developed a self-sustainable fever detector, which could be utilized for early detection and continuous monitoring of viral infectious diseases for humans and livestock. However, one-time measurements may suffer from false negative results because they could be strongly affected by human errors, environmental conditions, and stage of infection, in addition to a risk of cross infection. These drawbacks could be mitigated by continuous monitoring with low-cost and self-sustainable sensors and electrochromic display devices (Fig. S12 and Fig. 5c,d,e). To visualize the temperature changes, we integrated an electrochromic display with serially-connected four devices with the fully-developed oxidation layer. The electrochromic display was made of Prussian white (Fig. S13a). When electricity was supplied to the display, the color was changed from white to blue, and then, without electricity, from blue to white, reversibly. To mimic a situation with a fever, the heat flux from a human without and with a fever was assumed to be 360 W m^{-2} and 580 W m^{-2} , respectively, which have yielded the temperature differences (ΔT) of $1.8 \text{ }^\circ\text{C}$ and $2.9 \text{ }^\circ\text{C}$, respectively, across the device (Fig. 5a) (See Section 8 in SI)⁴²⁻⁴⁵. Under the temperature differences, voltages of $\sim 0.6 \text{ V}$ and $\sim 1 \text{ V}$

Deleted: battery-less and

Deleted: thermoelectric

were observed (Fig. 5b). Before the operation of the device, the current status window on the left in Fig. 5c is white. When ΔT was 1.8 °C, the color of the window was slightly changed to light blue, indicating the temperature is below a fever on the reference bar (Fig. 5d). Further increase in ΔT to 2.9 °C resulted in a darker blue window, indicating fever or higher temperature. For other or more sophisticated applications, the number of devices can be readily altered, and the color of the reference bar can be adjusted for desired temperature ranges.

In summary, we discovered a new method of generating an extremely large $\downarrow T$ toE factor, -87 mV K⁻¹ at 22 °C and 50% RH by utilizing the change in the corrosion potential due to the thermo-diffusion of water, through a series of systematic and rigorous experimental studies for unveiling the working mechanisms. We also substantiated the newly developed thermo-hydro-electrochemical conversion concept by powering electronic devices including a fever detector that can be distributed to many unspecified people at public places at an extremely low price. We anticipate that this study opens up and facilitates subsequent research for achieving even higher $\downarrow T$ toE factors as well as developing self-sustainable electronic devices including disposable, low-cost, and compact sensors.

Method

Materials and devices

Polystyrene sulfonic acid (PSS-H) (M.W. 75000, 30% wt%; Alfa Aesar), aniline (99+%; Alfa Aesar), hydrochloric acid (HCl, 36.5% - 38.0%, ACS; Macron Fine Chemicals), carbon steel shim (1008-1010 carbon steel, thickness: 0.005 inch; Precision Brand Products, Inc.), graphite foil (\geq

Deleted: thermopower

Deleted: our thermal energy harvester could be a viable option for

Deleted: battery-less

Deleted: thermopower

Deleted: nd

Deleted: such as

Deleted: due to battery-less operation

99.8% metals basis, thickness 0.254 mm; Alfa Aesar), iron(III) chloride (anhydrous; Sigma Aldrich), Iron(II) chloride (tetrahydrate, 98%; Alfa Aesar), iron foil (iron $\geq 99.99\%$ metals basis, thickness 0.1 mm; Alfa Aesar), ammonium persulfate (ACS; J.T. Baker), deionized (DI) water ($> 18 \text{ M}\Omega \text{ cm}^{-1}$), potassium ferricyanide ($\text{K}_3\text{Fe}(\text{CN})_6$, 98.5%; Acros Organics), tetraethylammonium perchlorate (TEAP, 98%; Alfa Aesar), ITO glass ($100 \Omega \text{ sq}^{-1}$; Nanocs), Nafion 115 membrane (Fuel Cell Earth), hydrazine (Anhydrous, 98%; Sigma Aldrich). The digital hygrometer with LCD screen was purchased from Linkstyle.

Material synthesis

The following procedure was used to synthesize the polyelectrolyte made of polyaniline and polystyrene sulfonate (PANI:PSS)³⁰. First, 30-wt% PSS-H (30 g) and aniline (4 mL) were added to DI water (80 mL), and the solution was stirred for 1 h. Ammonium persulfate solution was diluted in DI water (50 mg of ammonium persulfate per mL of water). Then, 40 mL of the aqueous ammonium persulfate solution was slowly dropped into the PANI:PSS solution for 30 min using a syringe pump while the solution was stirred using a magnetic bar. The solution was stirred for 5 min and then stored overnight. Subsequently, the solution was poured into acetone (1 L) for precipitation, and then decanted the solution. The collected precipitate were washed by acetone five times, and then dried at $50 \text{ }^\circ\text{C}$ in an oven to obtain dark green PANI:PSS (~13 g). The dried PANI:PSS precipitate dissolved in DI water (20 wt%) by stirring for 2 h, and then the hydrochloric acid was added to the PANI:PSS solution (10 vol% of the hydrochloric acid in the PANI:PSS solution). After the solution was stirred for 1 h, the solution (~3 g) was drop-casted on two carbon

steel foils (dimension is 2 cm by 2 cm), and two pieces were put together before they were fully dried (typically after 12 h in a fume hood). The sample was left in a fume hood for 5 days at room temperature. During the drying process, carbon paint was coated on the outer side of the carbon steel for good electrical connection with lead metal tab for electrical measurements. The thickness of the device was ~2 mm.

TtoE factor and electrochemical impedance spectroscopy (EIS) measurements

TtoE factor was measured using our custom-built setup (Fig. S3) in the humidity-controlled chamber. Peltier devices with aluminum block were used for controlling temperature. The device was placed in the middle of two Peltier devices to control the temperature difference. Two thermocouples were placed on top and bottom of the device to measure the temperature while two copper wires were used to measure the voltage between the two electrodes. Upon altering the current passing through the Peltier device, the temperature difference was varied. We took data after sample was left in a particular relative humidity level for at least 8 h to ensure the sample reached steady state. Voltage as a function of time was measured at various temperature differences, typically 6-8 points, and the linear slope in temperature difference (x-axis) vs. voltage (y-axis) was sought for finding the TtoE factor (flipped the sign like conventional thermopower). For EIS measurements, our device was kept under different humidity levels for 8 hours, and then it was scanned over a frequency range between 0.1~10⁶ Hz.

Deleted: Thermopower

Deleted: Thermopower

Deleted: thermopower

Deleted: The temperature difference was also created by one Peltier device as a heater and an aluminum block as a heat sink.
¶
Electrochemical impedance spectroscopy (EIS) measurement¶

Data availability

The data that support the findings of this study are available from the corresponding author upon reasonable request.

References

- 1 Mao, J. *et al.* Advances in thermoelectrics. *Adv. Phys.* **67**, 69-147 (2018).
- 2 Wang, H. & Yu, C. Organic thermoelectrics: materials preparation, performance optimization, and device integration. *Joule* **3**, 53-80 (2019).
- 3 Zhao, L.-D. *et al.* Ultrahigh power factor and thermoelectric performance in hole-doped single-crystal SnSe. *Science* **351**, 141-144 (2016).
- 4 He, W. *et al.* High thermoelectric performance in low-cost SnS_{0.91}Se_{0.09} crystals. *Science* **365**, 1418-1424 (2019).
- 5 Kang, S. D. & Snyder, G. J. Charge-transport model for conducting polymers. *Nat. Mater.* **16**, 252-257 (2017).
- 6 Kim, S. I. *et al.* Dense dislocation arrays embedded in grain boundaries for high-performance bulk thermoelectrics. *Science* **348**, 109-114 (2015).
- 7 Russ, B., Glauddell, A., Urban, J. J., Chabinye, M. L. & Segalman, R. A. Organic thermoelectric materials for energy harvesting and temperature control. *Nat. Rev. Mater.* **1**, 16050 (2016).
- 8 Sohn, A. & Yu, C. Ionic transport properties and their empirical correlations for thermal-to-electrical energy conversion. *Mater. Today Phys.* **19**, 100433 (2021).
- 9 Kim, S. L., Lin, H. T. & Yu, C. Thermally chargeable solid-state supercapacitor. *Adv. Energy Mater.* **6**, 1600546 (2016).
- 10 Akbar, Z. A., Jeon, J.-W. & Jang, S.-Y. Intrinsically self-healable, stretchable thermoelectric materials with a large ionic Seebeck effect. *Energy Environ. Sci.* **13**, 2915-2923 (2020).
- 11 Kim, S. L., Hsu, J.-H. & Yu, C. Intercalated graphene oxide for flexible and practically large thermoelectric voltage generation and simultaneous energy storage. *Nano Energy* **48**, 582-589 (2018).
- 12 Kim, S. L., Hsu, J.-H. & Yu, C. Thermoelectric effects in solid-state polyelectrolytes. *Org. Electron.* **54**, 231-236 (2018).
- 13 Jiao, F. *et al.* Ionic thermoelectric paper. *J. Mater. Chem. A* **5**, 16883-16888 (2017).
- 14 Kim, B., Hwang, J. U. & Kim, E. Chloride transport in conductive polymer films for an n-type thermoelectric platform. *Energy Environ. Sci.* **13**, 859-867 (2020).
- 15 Han, C.-G. *et al.* Giant thermopower of ionic gelatin near room temperature. *Science* **368**, 1091-1098 (2020).
- 16 Li, T. *et al.* Cellulose ionic conductors with high differential thermal voltage for low-grade heat harvesting. *Nat. Mater.* **18**, 608-613 (2019).

- 17 Xue, J. *et al.* Vapor-activated power generation on conductive polymer. *Adv. Funct. Mater.* **26**, 8784-8792 (2016).
- 18 Zhao, F., Liang, Y., Cheng, H., Jiang, L. & Qu, L. Highly efficient moisture-enabled electricity generation from graphene oxide frameworks. *Energy Environ. Sci.* **9**, 912-916 (2016).
- 19 Zhao, F., Wang, L., Zhao, Y., Qu, L. & Dai, L. Graphene oxide nanoribbon assembly toward moisture-powered information storage. *Adv. Mater.* **29**, 1604972 (2017).
- 20 Lee, S., Eun, J. & Jeon, S. Facile fabrication of a highly efficient moisture-driven power generator using laser-induced graphitization under ambient conditions. *Nano Energy* **68**, 104364 (2020).
- 21 Xu, T. *et al.* An efficient polymer moist-electric generator. *Energy Environ. Sci.* **12**, 972-978 (2019).
- 22 Liang, Y. *et al.* Electric power generation via asymmetric moisturizing of graphene oxide for flexible, printable and portable electronics. *Energy Environ. Sci.* **11**, 1730-1735 (2018).
- 23 Zhao, F., Cheng, H., Zhang, Z., Jiang, L. & Qu, L. Direct power generation from a graphene oxide film under moisture. *Adv. Mater.* **27**, 4351-4357 (2015).
- 24 Kim, B. *et al.* Robust high thermoelectric harvesting under a self-humidifying bilayer of metal organic framework and hydrogel layer. *Adv. Funct. Mater.* **29**, 1807549 (2019).
- 25 Choi, K., Kim, S. L., Yi, S.-i., Hsu, J.-H. & Yu, C. Promoting dual electronic and ionic transport in PEDOT by embedding carbon nanotubes for large thermoelectric responses. *ACS Appl. Mater. Interfaces* **10**, 23891-23899 (2018).
- 26 Kang, T. J. *et al.* Electrical power from nanotube and graphene electrochemical thermal energy harvesters. *Adv. Funct. Mater.* **22**, 477-489 (2012).
- 27 Lee, S. W. *et al.* An electrochemical system for efficiently harvesting low-grade heat energy. *Nat. Commun.* **5**, 3942 (2014).
- 28 Yu, B. *et al.* Thermosensitive crystallization–boosted liquid thermocells for low-grade heat harvesting. *Science*, **370**, 342 (2020).
- 29 Jang, J., Ha, J. & Cho, J. Fabrication of water-dispersible polyaniline-poly(4-styrenesulfonate) nanoparticles for inkjet-printed chemical-sensor applications. *Adv. Mater.* **19**, 1772-1775 (2007).
- 30 Mackay, A. L. β -Ferric oxyhydroxide. *Mineral Mag* **32**, 545-557 (1960).
- 31 Li, Y., Ying, B., Hong, L. & Yang, M. Water-soluble polyaniline and its composite with poly(vinyl alcohol) for humidity sensing. *Synth Met* **160**, 455-461 (2010).
- 32 Cheng, H., He, X., Fan, Z. & Ouyang, J. Flexible quasi-solid state ionogels with remarkable Seebeck coefficient and high thermoelectric properties. *Adv. Energy Mater.* **9**, 1901085 (2019).
- 33 Zhao, D. *et al.* Polymer gels with tunable ionic Seebeck coefficient for ultra-sensitive printed thermopiles. *Nat. Commun.* **10**, 1093 (2019).
- 34 Yang, P. *et al.* Wearable thermocells based on gel electrolytes for the utilization of body heat. *Angew. Chem. Int. Ed.* **128**, 12229-12232 (2016).
- 35 Bonetti, M., Nakamae, S., Roger, M. & Guenoun, P. Huge Seebeck coefficients in nonaqueous electrolytes. *J. Chem. Phys.* **134**, 114513 (2011).
- 36 Jiang, Q. *et al.* High thermoelectric performance in n-type perylene bisimide induced by the Soret effect. *Adv. Mater.*, 2002752 (2020).

- 37 Jeong, M. *et al.* Embedding aligned graphene oxides in polyelectrolytes to facilitate thermodiffusion of protons for high ionic thermoelectric figure-of-merit. *Adv Funct Mater* **n/a**, 2011016, (2021)
- 38 Cheng, F. *et al.* FTIR analysis of water structure and its influence on the flotation of arcanite (K_2SO_4) and epsomite ($MgSO_4 \cdot 7H_2O$). *Int. J. Miner. Process.* **122**, 36-42 (2013).
- 39 Stan, C. S., Popa, M., Olariu, M. & Secula, M. S. Synthesis and characterization of PSSA-polyaniline composite with an enhanced processability in thin films. *Open Chem.* **13**, 467-476 (2015).
- 40 Gardiner, C. & Melchers, R. Corrosion of mild steel in porous media. *Corros Sci* **44**, 2459-2478 (2002).
- 41 Yan, M., Sun, C., Xu, J. & Ke, W. Anoxic corrosion behavior of pipeline steel in acidic soils. *Ind. Eng. Chem. Res.* **53**, 17615-17624 (2014).
- 42 Stettler, N., Schutz, Y., Whitehead, R. & Jéquier, E. Effect of malaria and fever on energy metabolism in Gambian children. *Pediatr. Res.* **31**, 102-106 (1992).
- 43 Du Bois, E. F. The mechanism of heat loss and temperature regulation. *Ann. Intern. Med.* **12**, 338-395, (1938).
- 44 de Rivera, P. J. R., de Rivera, M. R., Socorro, F. & de Rivera, M. R. Measurement of human body surface heat flux using a calorimetric sensor. *J. Therm. Biol.* **81**, 178-184 (2019).
- 45 Herzog, L. W. & Coyne, L. J. What is fever? Normal temperature in infants less than 3 months old. *Clin Pediatr* **32**, 142-146 (1993).

Acknowledgement

The authors acknowledge financial supports from U.S. National Science Foundation (CBET 1805963).

Author contributions

C. Y. and Y. Z. conceived the idea, carried out the experiments and analyses, and wrote the manuscript. A. S. and A. C. assisted the experiments.

Ethics declarations

Competing interests

The authors declare no competing interests.

Figure 1. (a) Device fabrication procedure. Scanning electron micrographs of (b) the carbon steel electrode after detaching PANI:PSS, (c) a porous area, and (d) a flat area. (e) X-ray diffraction data for a pristine carbon steel, the electrode shown in 'b', and reference data for $\beta\text{-FeOOH}$ ³⁰. (f) Water uptake of PANI:PSS after the sample was left under the relative humidity for 8 h. Error bar: one standard deviation.

Figure 2. (a) Voltage profile of the device with the fully-developed oxidation layer under 50% RH as a function of time when the temperature difference (ΔT) was altered. The inset shows the saturated voltage at the corresponding temperature to seek the slope (93 mV K^{-1}). Note that the TtoE factor has the opposite sign (-93 mV K^{-1}) like conventional thermopower. (b) TtoE factors (absolute value) of our devices with the intermediate (IM) and fully-developed (FD) oxidation layers along with data in the literature, Kim^a, Han¹⁵, Kim^b, Jiao¹³, Akbar¹⁰, Kim^c, Kim^d, Cheng³², Kim^e, Kim^f, Kim^g, Zhao^a, Zhao^b, and Yang³⁴. Error bar: one standard deviation. ¹RH is assumed to be 50%.

Figure 3. (a) Experimental procedure to identify the influence of water uptake on the corrosion potential of carbon steel. (b) Voltage and change of water mass when the top electrode was left under different RH following the procedure in 'a'. (c) ATR-FTIR spectra of PANI:PSS from the colder side as the temperature difference (ΔT) was varied. The inset illustrates the in-situ ATR-FTIR experiment configuration. (d) Evans diagram shows the cathodic reduction reaction (black line), the anodic oxidation reaction (green line) under uniform temperature. The green line shifts towards the red (or blue) line with less (or more) water on the electrodes. (e-h) Illustration showing thermoelectric voltage generation and corresponding potential changes in the hotter and colder sides. (e) Uniformly distributed water in PANI:PSS under $\Delta T=0$. (f) As $\Delta T>0$, water molecules migrate from the hotter to colder side. (g) Voltage generation at steady-state under $\Delta T>0$. (h) Water molecules return to their initial distributed states under $\Delta T \rightarrow 0$.

Figure 4. (a) Voltage generation from one device under $\Delta T=8^\circ\text{C}$ followed by charging/discharging a capacitor. (b) Linearly increasing voltage with more number of devices connected in series under $\Delta T = 3.9$ and 7.6°C . The upper inset is the voltage profile as a function of time as 2, 4, 6, and 8 devices were joined in series. The lower inset depicts two devices connected in series. (c) Voltage and current profiles when the capacitor (470 μF) was hooked up to the digital hydrometer shown in the inset. (d) Voltage profiles for one module (m_1) and two modules connected in parallel (m_1 and m_2), followed by charging a capacitor (470 μF) as a function of time under $\Delta T = 7.6^\circ\text{C}$. Each module consists of 4 devices connected in series.

Figure 5. (a) Temperature difference as a function of time when heat is dissipated at the rates of 360 W m^{-2} and 580 W m^{-2} , to mimic normal and fever conditions, respectively. The inset illustrates the temperature measurement configuration with a heater. (b) Voltage profile as a function of time from serially connected four devices under the temperature differences of 1.8 and 2.9 °C. The inset illustrates the thermal energy harvester was directly connected to the fever detector. (c) A photograph of the fever detector with an electrochromic display window showing the current status and a reference color bar when heat is not supplied to the device ($\Delta T=0^\circ\text{C}$). (d) When $\Delta T \approx 1.8^\circ\text{C}$, the color in the status window became light blue, indicating the device was working under a normal condition. (e) When $\Delta T \approx 2.9^\circ\text{C}$, the status window displayed dark blue above the fever color in the reference bar.

SUPPLEMENTARY INFORMATION

Colossal Thermo-Hydro-Electrochemical Voltage Generation for Self-sustainable Operation of Electronics

Yufan Zhang¹, Ahrum Sohn², Anirban Chakraborty², Choongho Yu^{1,2} *

¹ Department of Materials Science and Engineering, Texas A&M University, College Station, Texas, 77843 USA

² Department of Mechanical Engineering, Texas A&M University, College Station, Texas, 77843 USA

*Corresponding author: chyu@tamu.edu

Deleted: Colossal Thermal-to-Electrical Energy Conversion in a Thermal-and-Chemical Energy Hybrid Device Thermopower for Battery-less and Self-sustainable Operation of Electronics ¶

1. Material Characterization

PANI:PSS was characterized by attenuated total reflectance (ATR) Fourier transform infrared spectroscopy (FTIR) (Thermo Nicolet 380 FTIR spectrometer), as shown in Fig. S1. The peaks at 1006 cm^{-1} and 1128 cm^{-1} correspond to benzene in-plane bending vibrations and skeleton vibrations, respectively. Out-of-plane deformation of C-H in the benzene ring is in the region of 823 cm^{-1} . Symmetric-stretching vibration of the sulfonic group attributes to the peak at 1034 cm^{-1} . The C-N stretching of the secondary aromatic amine appears at 1303 cm^{-1} . C-C stretching deformation of quinoid and C-C stretching deformation of benzenoid rings are at 1606 m^{-1} and 1490 cm^{-1} , respectively. These peaks match those in the literature ^{1,2}, which indicates that our PANI:PSS was synthesized successfully. The surface of electrodes was inspected by a JEOL JSM-7500F field emission scanning electron microscope. X-ray diffraction patterns were obtained using a Bruker D8 Discover X-ray diffractometer. Electrochemical impedance spectra was obtained by a Gamry Interface 1010 potentiostat, and Tafel curves were obtained using a CHI 604D electrochemical analyzer. The thermal conductivity of our whole device along the out-of-plane direction was measured to be $0.41\text{ W m}^{-1}\text{ K}^{-1}$ using the ASTM D5470 method.

Fig. S1. ATR-FTIR of PANI:PSS.

2. Water Uptake Measurement

Initially PANI:PSS was kept in an ambient condition (~22 °C) with 50 % relative humidity (RH). Then, PANI:PSS was placed in a humidity-controlled chamber, as described in our earlier work³. The mass of PANI:PSS was recorded every two hours. The water uptake after 8 h was saturated and plotted in Fig. 1f. Thermal-to-electrical conversion factor (TtoE factor) measurements were carried out after samples were left under the controlled humidity environment for 8 h. The water uptake was calculate as follows.

$$\text{Water uptake (\%)} = \frac{m_1 - m_{FD}}{m_{FD}} \times 100 \quad (1)$$

where m_1 is the total mass of PANI:PSS after it was exposed to an environment with different relative humidity. m_{FD} is the mass of fully dried PANI:PSS by evaporating water in a vacuum oven at 50 °C for 48 h.

Fig. S2. Water uptake in PANI:PSS as a function of time under 30%, 40%, 50%, 60% and 70% RH.

Deleted: Thermopower

3. TtoE factor Measurement

TtoE factors were measured using our custom-built setup (Fig. S3a), which is also described in our previous work³⁻⁵.

Fig. S3. (a) Experimental setup for TtoE factor measurements. Voltage vs. temperature difference under different relative humidity (RH) for devices with oxidation layers after (b) 14 days and (c) 60 days.

Deleted: hermopower

Deleted: hermopower

Deleted: was

Deleted: thermopower

Table S1. Slope and R^2 for the linear fit in Fig. S3b.

Relative humidity (% RH)	Slope (mV K ⁻¹)	R^2
30	27	0.9999
40	35	0.9987
50	47	0.9975
60	35	0.9956
70	16	0.9991

Table S2. Slope and R^2 for the linear fit in Fig. S3c.

Relative humidity (% RH)	Slope (mV K ⁻¹)	R^2
30	48	0.9985
40	63	0.9987
50	93	0.9962
60	65	0.9947
70	39	0.9945

4. Electrochemical Impedance Spectroscopy (EIS) Measurement

Nyquist plots are displayed in Fig. S4a, S4b, S4c, S4d and S4e. The impedance of the device decreased under higher relative humidity whereas the impedance increased under lower relative humidity. This implies that water uptake strongly affects the charge transfer. The semicircles indicating capacitive behaviors can be attributed to the electrically insulating FeOOH layer on carbon steel electrodes. Larger semicircles were observed from the device with the oxidation layer after 60 days than that of 14 days. This indicates that the impedance was increased due to more oxidation layers between the carbon steel electrodes and PANI:PSS. It should be noted that the oxidation layer was no longer developed further afterwards.

When graphite foil electrodes were used instead of carbon steel electrodes without the FeOOH layer, the impedance was significantly lowered, as depicted in Fig. S4f. We believe that the electrically insulating FeOOH layers plays an important role in creating large potential differences between two electrodes. The pores in FeOOH layer facilitates a large swing in water uptake, differentiating the potential of the two electrodes depending on the water uptake. Conversely, the graphite electrodes without the FeOOH layer is unfavorable to maintain a potential difference, if any, due to the relatively low impedance between the two graphite electrodes.

Fig. S4. (a) Nyquist plots from electrochemical impedance spectroscopy measurement of the device shown in Fig. 1 under (a) 50; (b) 30; (c) 40; (d) 60; (e) 70 %RH. (f) Nyquist plot of a test sample made of PANI:PSS and graphite electrodes instead of carbon steel electrodes.

5. Investigation of Working Mechanism

We have carried out systematic studies to unveil the working principle of our device. Rigorous literature survey and experiments brought us five different possible hypotheses as major causes for the voltage generation under temperature difference.

5.1. Ion concentration difference between the hotter and colder sides

The thermo-diffusion of ions in polyelectrolytes under a temperature gradient makes the ion concentration in the hotter and colder side different. Kim *et al.* described the relationship between proton concentration and voltage in PSS-H using the Nernst equation: ³

$$E_{cell} = E_{cell}^0 - \frac{RT}{F} \ln \frac{[H_{hot}^+]}{[H_{cold}^+]} \quad (2)$$

where E_{cell} is the voltage between hotter side and colder side; E_{cell}^0 is the standard cell potential from two electrodes; R is the universal gas constant; T is the temperature; F is the Faraday constant; and $[H_{hot}^+]$ and $[H_{cold}^+]$ are the proton concentration in the hotter side and colder side.

Fig. S5. (a) Device configuration of PANI:PSS as electrolyte and graphite foils as electrodes. (b) Voltage vs. temperature difference plot with the trend line for measuring TtoE factor. The slope is negative, so the TtoE factor is positive like conventional thermopower.

Deleted: thermopower

Deleted: thermopower

To investigate the influence of the ion concentration difference on the TtoE factor, carbon steel electrodes were substituted by graphite foils so as to eliminate the corrosion effect from carbon steel, as illustrated in Fig. S5a. Based on the relationship between temperature difference and voltage under the ambient condition, a negative slope of -0.97 mV K^{-1} from the linear fitting was obtained. When a temperature difference is created, protons migrates from the hotter to the colder side, as illustrated in the inset of Fig. S5b. Then the colder side has a higher potential due to a higher concentration of positive ions, resulting in a negative slope. This trend is opposite to that of our PANI:PSS (see the inset of Fig. 2a). Furthermore, the absolute value of the slope is much smaller than the TtoE factor of our device. Therefore, we concluded that the thermo-diffusion of ions is not the decisive contributor of the TtoE factor observed in our carbon steel based device shown in Fig. 1a.

Deleted: thermopower

Deleted: thermopower

Deleted: thermopower

5.2. Comparative study about the working principles of our devices and moisture-powered devices

Moisture-powered devices generate voltage by carrying ions (protons) during water diffusion through a hygroscopic medium and thereby inducing charge imbalance when one of the two sides in the medium is exposed to more water^{6,7}. In literature, metal and carbon electrodes (e.g., gold, graphite) were used together with a hygroscopic materials in between^{7,8}. In our work, we assembled two devices with carbon steel or graphite foil electrodes in Fig. S6a or Fig. S6b, respectively. We sealed the side walls with silicone to ensure that water pass through PANI:PSS without evaporation from the side walls. When the devices initially at a 27% RH environment were moved to a 85% RH environment, the voltage from the device with carbon steel electrodes was changed to a large negative voltage, -360 mV (Fig. S6c) whereas that of the graphite foil

electrodes displayed less than +1 mV (Fig. S6d). This result shows that ion migration due to water diffusion is not a major contributor to the voltage generation in our devices unlike the moisture-powered devices. Instead, more water near the hole side gave rise to more negative potential than the other side.

Fig. S6. (a) Device configuration of PANI:PSS as an electrolyte and carbon steels with holes and without holes as electrodes. (b) Device configuration of PANI:PSS as an electrolyte and graphite foils with holes and without hole as electrodes. Voltage response as a function of time under humidity change in the devices with (c) carbon steel electrodes and (d) graphite electrodes.

5.3. Difference in the redox potential between the hotter and colder sides

We also tested if the thermoelectric voltage in our device can be attributed to the redox reaction of $\text{FeOOH}/\text{Fe}^{2+}$, which is described as follows.

We designed an experiment using FeOOH products from the carbon steel electrode of our device in a configuration depicted in Fig. S7a. To attach the FeOOH products to the graphite foils, the FeOOH products were spread on the graphite foils and two drops of a liquid-type superglue were applied and subsequently more FeOOH products were spread on top of the first layer. Then PANI:PSS was sandwiched between the two graphite electrodes. In this configuration, FeOOH and Fe^{2+} are provided by corrosive products and H^+ and H_2O are from polyelectrolyte. To investigate the temperature effect of this redox reaction, voltage change was recorded when temperature gradients were created. Fig. S7b displays varying voltage responses at a constant temperature difference with small thermoelectric voltage. This result indicates that temperature-dependent redox reactions from FeOOH/ Fe^{2+} are not the major cause for the colossal TtoE factor observed in our carbon steel based device shown in Fig. 1a.

Deleted: thermopower

Fig. S7. (a) A testing device configured for temperature-dependent redox reactions. Corrosive FeOOH products were sandwiched between a graphite foil electrode and PANI:PSS. (b) Voltage response as a function of time under a temperature difference.

5.4. Variation of corrosion potential due to the change in the ion concentration

When protons diffuse under a temperature gradient, it is possible to change the corrosion potential like pH-dependent corrosion of carbon steel^{9,10}. Here, we designed an experiment to find out the relationship between pH and the corrosion potential of carbon steel. Tafel curve is one of the most effective method to measure the corrosion potential¹¹.

Fig. S8. (a) Three electrodes configuration for the measurement of the Tafel curve while 1M HCl was added to the solution. (b) Tafel curves for the carbon steel electrode taken out of our device in the solutions at different pH.

Figure S8a depicts our experimental setup consisting of a working electrode from our device (shown in Fig. 1), Pt counter electrode, and Ag/AgCl reference electrode in an aqueous 1 wt% PANI:PSS solution. Initially, pH was measured to be 1.25. Tafel curves were recorded as 1M HCl was added to have pH=1, 0.8, and 0.6, as displayed in Fig. S8b. When the current between the working and counter electrode became small, the corrosion potential between the working and reference electrodes was changed from -0.45 V to -0.39 V with diminishing pH (*i.e.*, increasing proton concentration). The potential gets higher with more protons, which is similar to the inset of Fig. S5b. This denotes our device in the main manuscript operates under a different working principle.

We designed another experiment with 1M FeCl₃ solution as an electrolyte and the carbon steel electrodes detached from the device as electrodes, as illustrated in Fig. S9a since Fe ions participate in the corrosion process. Voltage was measured as concentrated HCl (12M) was added to the solution where the positive terminal was immersed. Figure S9b displays positively increasing voltage with more protons, denoting that the potential is higher with more protons like the trends shown in Fig. S8 and Fig. S5. Therefore, we excluded this as a major contributor for the TtoE factor in our device.

Deleted: thermopower

Fig. S9. (a) Experimental setup for measuring voltage as HCl was gradually added. (b) Voltage profile as a function of time when HCl was added to the solution where the carbon steel electrode connected to the positive terminal was immersed.

5.5. Corrosion potential due to the change in temperature

The experimental setup in Fig. S8a was used again, but at this time the temperature of the entire beaker was immersed in a water bath instead of HCl addition, as illustrated in Fig. S10a. The water bath was used for facilitating uniform temperature of the PANI:PSS solution during the experiment. The corrosion potentials of the electrode at different temperatures are displayed in the Tafel curve (Fig. 10b).

Fig. S10. (a) Three electrode configuration for the measurement of the Tafel curves at different temperatures. (b) Tafel curves for the carbon steel electrode detached from our device when the solution temperature was varied.

We observed only 7 mV difference while the temperature changed from 15 °C to 40 °C. When the temperature was raised, the corrosion potential was lowered. This trend is also different from our device. Temperature effect on the corrosion potential is much smaller than the colossal TtoE factor from our device, suggesting temperature variation is also not the major driver of our system.

Deleted: thermopower

5.6. Corrosion potential due to the change of water mass in the electrodes

Carbon steel corrodes when an electrochemical cell is formed on its surface. Figure S11a shows a cross-section of a carbon steel electrode after 6 months. The thickness of the carbon steel has been reduced from $\sim 125 \mu\text{m}$ to $\sim 107 \mu\text{m}$ after 6 months. The average reduction in thickness assuming a constant corrosion rate (which is in fact not constant) is only $\sim 3 \mu\text{m month}^{-1}$. Four essential elements for this electrochemical cell for corrosion include (a) anodic reaction, (b) cathodic reaction, (c) metallic path between anodic and cathodic sites, and (d) electrolyte, as illustrated in Fig. S11b. Anodic and cathodic sites are in carbon steel surface, and iron metal connects these two sites so that electrons can pass through. Water is the most common electrolyte for carbon steel corrosion. Yan et al. studied how the amount of water in a soil environment (carbon steel buried in soil) affects carbon steel corrosion¹². This study delineated that water uptake less than 30% did not cover all the surfaces of the carbon steel. Then electrochemical cells on the surface of the carbon steel would form in the area where water was present. Water is the electrolyte so corrosion is strongly dependent on the amount of water at the local site, which can be described by the overpotential ($E_{\text{overpotential}} = I_i R_i$, where I_i is the corrosion current and R_i is the resistance of electrolyte). With more water, the resistance becomes smaller so the overpotential becomes smaller and vice versa, as described in the main manuscript along with Fig. 3d. Water in PANI:PSS diffuses upon imposing a temperature gradient. Then, the amounts of water in the two carbon steel electrodes become different, resulting in dissimilar overpotentials, and thereby nonzero voltage between the two electrodes are observed.

Fig. S11. (a) SEM image of a carbon steel electrode (cross section) after 6 months. The thickness of the electrode was reduced to $\sim 107 \mu\text{m}$ from the initial thickness of $\sim 125 \mu\text{m}$, showing $\sim 3 \mu\text{m month}^{-1}$ reduction in thickness. (b) Illustration of the carbon steel electrode where FeOOH covered its surface with water in between. (c) Mass change in a single carbon steel electrode after the sample was left under different relative humidity for 12 h.

6. In-situ ATR-FTIR for Investigating Thermo-diffusion of Water

In-situ ATR-FTIR was carried out with a configuration in the inset of Fig. 3c. PANI:PSS was placed on the crystal of ATR accessory and a heater was placed on top of the PANI:PSS sample. The heater was pressed from the top to ensure PANI:PSS makes good contacts with the heater and the crystal surface. The first scan was carried out at room temperature in the range of 4000~2000 cm^{-1} without turning on the heater. Subsequently, the heater was turned on. After about 30 seconds, the scanning began and, after about 90 seconds, it was finished. The thermocouples recorded the temperature (T_1 and T_2) at the end of the scan. Then the power of the heater was raised, and the ATR-FTIR and temperature data were recorded. This procedure was repeated 4 times to obtain the FTIR absorbance spectra as a function of temperature difference. Table S3 includes temperature measurement results.

Table S3. Temperature during the in-situ ATR-FTIR measurement.

Time (min)	T_2 ($^{\circ}\text{C}$)	T_1 ($^{\circ}\text{C}$)	Temperature difference ($^{\circ}\text{C}$)
0	23.6	23.6	0.0
2	25.8	24.3	1.5
4	27.6	24.5	3.1
6	29.8	24.8	5.0
8	32.5	25.0	7.5

The peak at 2914 cm^{-1} corresponding to C-H stretching in CH_2 group of PSS was used to normalize FTIR spectra for better comparison. The O-H stretching band from water is broad over around 2800~3700 cm^{-1} , the intensity of nearby peaks at 3340 and 3240 cm^{-1} , corresponding to N-H stretching, could be affected by the change of the O-H peak.

7. Operation of Electrochromic Devices

Figure S12a shows an electrochromic device (Ynvisible Interactive Inc.) without any electrical connection. According to the specification, this electrochromic device shows patterns when electricity is supplied with a voltage of ~ 1.5 V. When the electrochromic device was hooked up with our serially connected four devices under a temperature difference of 5 $^{\circ}\text{C}$, the pattern has been changed (Fig. S12b), indicating that electricity was supplied to the electrochromic device. When the positive and negative terminals were switched, the pattern was reversed (Fig. S12c).

Fig. S12. Photographs of an electrochromic device (a) without any electrical connection, (b) after connecting our devices (four in series) under a temperature difference of 5 $^{\circ}\text{C}$, and (c) after flipping the positive and negative terminals.

8. Self-sustainable Fever Detection

We have fabricated a fever indicator based on electrochromic color changes by following our earlier work⁵. The structure of our fever indicator is shown in Fig. S13a. First, 0.0052-M HCl, 0.01-M $\text{K}_3\text{Fe}(\text{CN})_6$ and 0.01-M FeCl_3 solution in DI water was prepared. A prussian blue (PB) film was deposited on a ITO glass by a cyclic voltammetry method from 0 to 2 V with 0.05 V s^{-1}

scan rate for 3 cycles in a 50-mL solution. We used a tape-masked ITO glass as the working electrode, Ag/AgCl as the reference electrode, and Pt as the counter electrode. Then, the PB film on the ITO glass was placed in a closed chamber where hydrazine was vaporized for 3 mins in order to reduce PB to prussian white (PW). To fabricate the electrochromic device, a Nafion 115 membrane was pre-soaked in a 0.125-M TEAP solution and then sandwiched between the ITO glass with PW and another bare ITO glass. Finally, epoxy was used to seal the device to avoid the evaporation of the TEAP electrolyte.

We have estimated operating conditions when our energy harvesting device was used for those who have a fever. The heat dissipation from the humans without fever was found to be 200–500 W m⁻² ¹³, so we assumed 360 W m⁻² is a normal condition. In general, a body temperature above 38.1 °C is considered to be a fever ¹⁴. It has been reported that ~13% increase in metabolism per 1 °C rise in the temperature of a human body ^{15,16}. Considering the maximum heat flux from healthy humans is 500 W m⁻², we assumed that the heat flux from humans with fever is 580 W m⁻². These values could be adjusted, if necessary, depending on needs and other considerations.

Fig. S13. (a) Schematic illustration of the fever indicator. (b) Temperature rise of T_1 and T_2 from the initial temperature in our energy harvesting device as a function of time under two different heat flux conditions.

References

- 1 Li, Y., Ying, B., Hong, L. & Yang, M. Water-soluble polyaniline and its composite with poly (vinyl alcohol) for humidity sensing. *Synth Met* **160**, 455-461 (2010).
- 2 Stan, C. S., Popa, M., Olariu, M. & Secula, M. S. Synthesis and characterization of PSSA-polyaniline composite with an enhanced processability in thin films. *Open Chem.* **13**, 467-476, (2015).
- 3 Kim, S. L., Lin, H. T. & Yu, C. Thermally chargeable solid-state supercapacitor. *Adv. Energy Mater.* **6**, 1600546, (2016).
- 4 Kim, S. L., Hsu, J.-H. & Yu, C. Thermoelectric effects in solid-state polyelectrolytes. *Org. Electron.* **54**, 231-236, (2018).
- 5 Kim, S. L., Hsu, J.-H. & Yu, C. Intercalated graphene oxide for flexible and practically large thermoelectric voltage generation and simultaneous energy storage. *Nano Energy* **48**, 582-589, (2018).
- 6 Zhao, F., Wang, L., Zhao, Y., Qu, L. & Dai, L. Graphene Oxide Nanoribbon Assembly toward Moisture-Powered Information Storage. *Adv. Mater.* **29**, 1604972, (2017).
- 7 Xu, T. *et al.* An efficient polymer moist-electric generator. *Energy Environ. Sci.* **12**, 972-978, (2019).
- 8 Liang, Y. *et al.* Electric power generation via asymmetric moisturizing of graphene oxide for flexible, printable and portable electronics. *Energy Environ. Sci.* **11**, 1730-1735, (2018).
- 9 Pessu, F., Barker, R. & Neville, A. The influence of pH on localized corrosion behavior of X65 carbon steel in CO₂-saturated brines. *Corrosion* **71**, 1452-1466, (2015).
- 10 Zheng, Y., Ning, J., Brown, B. & Nešić, S. Electrochemical model of mild steel corrosion in a mixed H₂S/CO₂ aqueous environment in the absence of protective corrosion product layers. *Corrosion* **71**, 316-325, (2014).
- 11 McCafferty, E. *Introduction to Corrosion Science.* (Springer New York, 2010).
- 12 Yan, M., Sun, C., Xu, J. & Ke, W. Anoxic corrosion behavior of pipeline steel in acidic soils. *Ind Eng Chem Res* **53**, 17615-17624 (2014).
- 13 de Rivera, P. J. R., de Rivera, M. R., Socorro, F. & de Rivera, M. R. Measurement of human body surface heat flux using a calorimetric sensor. *J. Therm. Biol.* **81**, 178-184, (2019).
- 14 Herzog, L. W. & Coyne, L. J. What is fever? Normal temperature in infants less than 3 months old. *Clin Pediatr* **32**, 142-146 (1993).
- 15 Stettler, N., Schutz, Y., Whitehead, R. & Jéquier, E. Effect of malaria and fever on energy metabolism in Gambian children. *Pediatr. Res.* **31**, 102-106, (1992).
- 16 Du Bois, E. F. The mechanism of heat loss and temperature regulation. *Ann. Intern. Med.* **12**, 338-395, (1938).